# Urinary microRNAome in healthy cats and cats with pyelonephritis or other urological conditions

Marta Gòdia[1,2¤], Louise Brogaard[3☯], Emilio Mármol-Sánchez[4,5☯], Rebecca Langhorn[6], Ida Nordang Kieler[6], Bert Jan Reezigt[7], Lise Nikolic Nielsen[6], Lisbeth Rem Jessen[6‡*], Susanna Cirera[3‡*]

1 Department of Animal Medicine and Surgery, School of Veterinary Sciences, Universitat Autònoma de Barcelona, Cerdanyola del Vallès, Catalonia, Spain, 2 Centre for Research in Agricultural Genomics (CRAG) CSIC-IRTA-UAB-UB, Campus UAB, Cerdanyola del Vallès, Catalonia, Spain, 3 Department of Veterinary and Animal Sciences, Faculty of Health and Medical Sciences, University of Copenhagen, Frederiksberg, Denmark, 4 Department of Molecular Biosciences, The Wenner-Gren Institute, Stockholm University, Stockholm, Sweden, 5 Centre for Paleogenetics, Stockholm University, Stockholm, Sweden, 6 Faculty of Health and Medical Sciences, Department of Veterinary Clinical Sciences, University of Copenhagen, Frederiksberg, Denmark, 7 Blue Star Small Animal Hospital, Gothenburg, Sweden

☯ These authors contributed equally to this work.
¤ Current address: Animal Breeding and Genomics, Wageningen University and Research, Wageningen, The Netherlands
‡ These authors also contributed equally to this work.
* lrmj@sund.ku.dk (LRJ); scs@sund.ku.dk (SC)

**Data Availability Statement:** The datasets generated and analyzed in the current study are available at NCBI's BioProject PRJNA788683.

## Abstract

MicroRNAs (miRNAs) are short non-coding RNAs that regulate gene expression at the post-transcriptional level. miRNAs have been found in urine and have shown diagnostic potential in human nephropathies. Here, we aimed to characterize, for the first time, the feline urinary miRNAome and explore the use of urinary miRNA profiles as non-invasive bio-markers for feline pyelonephritis (PN). Thirty-eight cats were included in a prospective case-control study and classified in five groups: healthy Control cats (n = 11), cats with PN (n = 10), cats with subclinical bacteriuria or cystitis (SB/C, n = 5), cats with ureteral obstruction (n = 7) and cats with chronic kidney disease (n = 5). By small RNA sequencing we identified 212 miRNAs in cat urine, including annotated (n = 137) and putative novel (n = 75) miRNAs. The 15 most highly abundant urinary miRNAs accounted for nearly 71% of all detected miRNAs, most of which were previously identified in feline kidney. Ninety-nine differentially abundant (DA) miRNAs were identified when comparing Control cats to cats with urological conditions and 102 DA miRNAs when comparing PN to other urological conditions. Tissue clustering analysis revealed that the majority of urine samples clustered close to kidney, which confirm the likely cellular origin of the secreted urinary miRNAs. Relevant DA miRNAs were verified by quantitative real-time PCR (qPCR). Eighteen miRNAs discriminated Control cats from cats with a urological condition. Of those, seven miRNAs were DA by both RNA-seq and qPCR methods between Control and PN cats (miR-125b-5p, miR-27a-3p, miR-21-5p, miR-27b-3p, miR-125a-5p, miR-17-5p and miR-23a-3p) or DA between Control and SB/C cats (miR-125b-5p). Six additional miRNAs (miR-30b-5p, miR-30c, miR-30e-5p, miR-27a-3p, miR-27b-39 and miR-222) relevant for discriminating PN from other urological

**Funding:** LRJ, SC, LNN and BJR received funding from the AGRIA research foundation, nr. N2018-0007.

**Competing interests:** The authors have declared that no competing interests exist.

conditions were identified by qPCR alone (n = 4) or by both methods (n = 2) ($P$<0.05). This panel of 13 miRNAs has potential as non-invasive urinary biomarkers for diagnostic of PN and other urological conditions in cats.

## Introduction

MicroRNAs (miRNAs) are short non-coding RNAs that play an important role in gene regulation by binding to the 3'UTR of targeted mRNAs and triggering their degradation and/or inhibition of translation [1]. Although the cat genome was first sequenced in 2005 [2], the feline miRNAome has not yet been extensively annotated. In fact, there are no feline miRNAs present in miRBase [3], miRGeneDB [4] or miRCarta [5], the three most widely used miRNA databases. Nevertheless, the current genome assembly annotation of the domestic cat (Felis_catus_9.0) includes around 200 annotated feline miRNAs (www.ensembl.org), and recent studies profiling the feline miRNAome included their characterization in a feline kidney cell line [6] and *de novo* miRNA annotation in 12 different healthy cat tissues with high-throughput sequencing [7]. However, novel cat-specific miRNAs are still expected to be found in newly studied feline tissues/fluids due to their expected tissue specificity [8].

Many miRNAs exert an important role in the pathophysiology of human disease [9]. Specifically for the urinary tract, kidney miRNA expression profiles have been studied as putative biomarkers of acute kidney injury and chronic kidney disease [10,11]. For instance, some miRNAs have shown discriminatory potential in the diagnosis of acute allograft pyelonephritis [12]. miRNAs can be released into body fluids through diverse mechanisms, such as secretion of miRNA-containing micro-vesicles or apoptotic bodies [13]. Because of this, recent studies have been exploring urine as a convenient, non-invasively obtained surrogate for kidney tissue. The presence of miRNAs in body fluids has been shown to specifically relate to their associated tissues; as such, urinary miRNA levels showed the strongest correlation to kidney miRNA expression levels in a study that queried 40 different human tissues [14]. Moreover, abnormal expression of urinary miRNAs has been identified in humans with various renal pathologies [15].

The differential abundance of urinary miRNAs has been previously described in cats with urological diseases such as chronic kidney disease (CKD) [16] and, recently, pyelonephritis (PN) [17]. PN is an infection of the renal pelvis most commonly caused by ascending bacteria from the lower urinary tract. This condition represents a diagnostic challenge in cats, as the clinical picture may be very unspecific. Definitive diagnosis requires a positive urine culture obtained from the renal pelvis by pyelocentesis, an invasive and technically difficult procedure. In general veterinary practice, diagnosis is often tentative, based on a positive urine culture from the bladder in cats with clinical, laboratory, and/or ultrasonographic evidence indicating upper urinary tract infection. However, clinical, laboratory, and ultrasonographic features may overlap between cats with PN and cats with other urological conditions such as CKD, ureteral obstruction (UO) or lower urinary tract infections (i.e. subclinical bacteriuria (SB) or cystitis (C)) [18]. In addition, multiple disease processes may coexist in one animal [19,20]. Accurate identification of PN is of utmost importance, as failure to treat the infection may compromise renal function [21]. On the other hand, antimicrobial administration to cats with CKD, UO or SB without PN is inappropriate [22], and treatment of bacterial cystitis requires a different antimicrobial regime, stressing the need for discriminative diagnostic biomarkers.

We have recently conducted a prospective case-control study [17], assessing the stability of miRNAs in feline urine and the detectability of 24 pre-selected miRNAs in cats with PN and

other urological conditions. Increased levels of urinary miR-16 were detected in cats with PN compared to healthy cats and cats with other urological conditions, and clustering of miRNAs from cats with *E.coli* infections was noted [17]. Our results suggested that urinary miRNAs might be useful diagnostic biomarkers in feline urinary tract disease, thus justifying additional exploration of the feline urinary miRNAome in this population.

The aim of the present study was to (i) characterize the feline urinary miRNAome by profiling miRNAs in urine using small RNA sequencing and (ii) identify a panel of differentially abundant urinary miRNAs in cats with PN and cats with other urological diseases using qPCR, in order to establish discriminatory abundance profiles for diagnostic use.

## Materials and methods

The current study was designed as a case-control study and performed on urine samples from cats prospectively enrolled, as well as on urine samples stored from cats enrolled in a previous prospective study described in Jessen *et al.* 2020 [17].

### Recruitment and classification of cats

Cats were recruited from the University Hospital for Companion Animals in Copenhagen from 2015 to 2019 (UHCA) and the Blue Star Animal Hospital in Gothenburg (BSAH) from 2015 to 2017. Informed owner consent was obtained from all cat owners at the time of enrolment. Ethics approval was provided (nr. 2019–17) by the Administrative and Ethical Committee at the Department of Veterinary Clinical Sciences, University of Copenhagen.

Cats were classified into three categories: (i) healthy Control cats, (ii) cats with PN, and (iii) cats with other urological conditions (Table 1). Cats were classified as healthy based on

**Table 1. Demographic and microbiological data of included cats.**

| | Control | PN | SB/C | UO | CKD |
|---|---|---|---|---|---|
| **Number of cats** | 11 | 10 | 5 | 7 | 5 |
| **Age in years (median (range))** | 6 (2–16) | 7 (2–13) | 7 (1–10) | 5 (2–11) | 12 (8–17) |
| **Breed** | 8 DSH, 2 Ragdolls, 1 NFC | 5 DSH, 2 Birmans, 1 Abyssinian, 1 MC, 1 Oriental | 3 DSH, 2 MC | 2 DSH, 1 Abyssinian, 1 Birman, 1 Oriental, 1 MB, 1 Ocicat | 3 DSH, 2 DLH |
| **Sex** | 2 FE, 4 FN, 2 ME, 3 MN | 3 FE, 6 FN, 1 MN | 2 FE, 2 FN, 1 MN | 3 FN, 4 MN | 1 FE, 1 FN, 3 MN |
| **Uropathogens** | Culture-negative cystocentesis | Pyelocentesis: 5 *E. coli* 1 *Staph. pseudointermedius* 1 *Staph. felis* 1 *Enterococcus fecalis* (this cat had *E. coli* in urine cultured by cystocentesis) Cystocentesis: 2 *E. coli* | Cystocentesis: 2 *E. coli* 1 *Micrococcus luteus* 1 *Staph. felis* 1 *Streptococcus spp* | Culture-negative pyelocentesis | Culture-negative cystocentesis |
| **Comorbidities/ staging** | - | 1 CKD IRIS stage 1 (renal cyst) 1 non-obstructive nephrolithiasis 1 bilateral UO and evidence of CKD (unstaged) 2 unilateral UO 3 unilateral UO and evidence of CKD (unstaged) | 1 CKD IRIS stage III 1 likely CKD stage 1 and cystolithiasis | 4 evidence of CKD (unstaged) | 3 IRIS stage II 1 IRIS stage III 1 IRIS stage IV |

Of the selected population, 11 healthy Control, 5 PN, 3 SB/C, 5 UO, and 5 CKD cats have been previously described in Jessen *et al.* 2020 [17].

CKD: Chronic kidney disease, PN: Pyelonephritis, SB/C: Subclinical bacteriuria/Cystitis, UO: Ureteral obstruction, DSH: Domestic Shorthair, DLH: Domestic Longhair, NFC: Norwegian Forest Cat, MC: Maine Coon, MB: Mixed breed, FE: Female entire, FN: Female neutered, ME: Male entire, MN: Male neutered, *Staph*: *Staphylococcus*, IRIS: International Renal Interest Society [23].

unremarkable clinical and laboratory findings. Cats were classified as PN based on either a positive pelvic urine culture or a clinical presentation highly suggestive of PN, defined as presence of pyrexia, renal pain, azotemia and/or elevated acute phase proteins, along with a positive bladder urine culture. Cats fulfilling those criteria were classified as PN regardless of whether additional urological comorbidities were present or not. Cats were classified as having a urological condition other than PN if diagnosed with either UO, CKD, SB/C, or a combination thereof. The UO category included cats with uni- or bilateral ureteral obstructive disease and negative pelvic urine bacterial culture(s). The CKD category included cats with stable azotemic renal disease (based on International Renal Interest Society (IRIS) stage $\geq$ 2) [23] and negative bladder urine culture. The SB/C category included cats with a positive bladder urine culture and absence of clinical signs suggestive of PN. In these cats, clinical signs of lower urinary tract infection including pollakiuria, dysuria, stranguria, and/or hematuria were either absent (subclinical bacteriuria, SB) or present (cystitis, C).

## Diagnostic work-up

The protocol for diagnostic work-up was identical to the one described in Jessen *et al.* 2020 [17]. In brief, all cats included in the current study had basic work-up performed consisting of clinical examination, hematology, biochemistry, urinalysis, and urine bacterial culture and susceptibility testing (C&S).

In addition, cats with PN and cats with UO all had abdominal ultrasonography performed, followed by pyelocentesis and bacterial culture of pelvic urine in cats with abnormal dilation (> 2 mm) of the renal pelvis. Other procedures performed for diagnostic purposes in individual cats included abdominal radiography, antegrade pyelogram and/or IV excretory pyelogram, blood pressure measurement, blood symmetric dimethylarginine and total thyroxin measurements.

## Sampling, processing and storage of urine

Whole urine samples for urinalysis and for miRNA sequencing were collected from the bladder by cystocentesis. Urine for C&S testing was obtained either by cystocentesis and/or by pyelocentesis (cats with PN and UO). Urinalysis was performed within 30 minutes of sampling and the remaining urine aliquoted for bacterial culture and RNA isolation.

Urine was cultured by inoculation on 5% calf blood agar plates and incubated at 37˚C for 24 hours. Colony types were identified to the species level by matrix-assisted laser desorption/ionization time of flight (MALDI-TOF) mass spectrometry (Vitek MS RUO, BioMérieux). Antimicrobial susceptibility was tested for all isolates using the broth microdilution method (Sensititre® COMPAN1F, TREK Diagnostic System Ltd.) according to the Clinical and Laboratory Standards Institute [24].

For RNA isolation, aliquots of whole urine were frozen at -80˚C within a maximum of 2 hours after sampling unless collected during closing hours. Samples collected during closing hours were refrigerated and subsequently frozen at -80˚C within the following 24 hours. Urine samples from other labs were stored on site at -20˚C and transported on dry ice (-80˚C), batched and stored for a maximum of 4 years.

## RNA extraction, library preparation and sequencing

RNA from 200 μL of whole urine was extracted using the miRNeasy Mini Kit (Qiagen) according to the manufacturer's protocol. Briefly, five volumes of Qiazol were mixed with 200 μL of urine. The samples were then incubated for 5 minutes at room temperature and, subsequently, 200 μL chloroform were added. From here, manufacturer's instructions were followed as

specified, and RNA was finally eluted in 30 μL of RNase free water. RNA purity and concentration was assessed using a NanoDrop 2000 spectrophotometer (Thermo Fisher Scientific). Six μL of RNA from each sample (n = 38) were supplied to the NGS Service Provider (Genomics Unit, Center for Genomic Regulation, Barcelona, Spain). Small RNAseq libraries were prepared with the NEBNext® Small RNA Library Prep kit (New England Biolabs). The resulting 38 small RNA libraries were sequenced on an Illumina HiSeq 2500 system to generate 50 bp single-end reads.

## Read quality assessment and mapping to the feline genome

The bioinformatics workflow is provided in S1 Fig. Briefly, raw sequenced reads were analyzed for initial quality check with the FastQC v.0.11.5 software (http://www.bioinformatics. babraham.ac.uk./projects/fastqc/). Reads with low quality (phred–Q 20) and short length (<10 bp) were discarded. Primer adaptors were removed using Cutadapt v.1.0 [25]. Quality-check-trimmed reads were then mapped to the feline genome (Felis_catus_9.0) using the Bowtie v.1.2.3 aligner [26] with default parameters, except for "-l 25" (i.e. allowing a maximum of two mismatches within the first 25 nts of the read). Reads with up to 50 multiple alignments ("-m 50") were considered valid, and only the first single best stratum alignment ("—best—strata") was reported.

## Identification of known and novel miRNAs in feline urine

To date, feline miRNAs have not been annotated in any of the reference miRNA databases [3–5]. Nevertheless, there are two different sources of annotated feline miRNAs: (i) the Ensembl (v.99) database and (ii) the feline miRNAome from Laganà *et al.* 2017 [7], which queried 12 different healthy tissues (comprising lip, brain, tongue, testis, ovary, liver, pancreas, kidney, lymph node, skin, spleen and lung) by small RNA sequencing. In this study, we took advantage of these previously annotated miRNAs in cats, but also performed a *de novo* miRNA prediction analysis as urine miRNAs are expected to be highly tissue-specific [8].

In order to identify known and putative novel feline miRNAs, all non-miRNA reads were first filtered out (S1 Fig). For this, reads that mapped to the feline genome were then compared with Silva [27], GtRNAdb [28], RepeatMasker [29], and non-coding Ensembl (v.99) databases using Bowtie v.1.2.3 [26], in order to remove reads mapping to ribosomal RNA (rRNA), transfer RNA (tRNA), small nucleolar RNA (snoRNA), small nuclear RNA (snRNA), and repeat elements (REs). After this initial filtering process, the remaining reads with sequence length between 18 and 30 nts were kept and used for the identification of known miRNAs and further prediction of putative novel miRNAs.

First, reads were queried against the known annotated miRNAs. This includes 218 feline miRNAs from the Ensembl database and the 271 miRNAs from the feline miRNAome of Laganà *et al.* 2017 [7]. Reads that remained unmapped to any of these known miRNAs were then used for the *de novo* prediction of putative miRNA candidates by using the miRDeep2 software [30]. The miRDeep2 algorithm uses Bayesian statistics to score the likelihood of sequenced RNAs to belong to the established model of miRNA biogenesis [30]. The software also makes use of the RNAfold tool from the ViennaRNA package 2.0 [31], which predicts the RNA secondary structure of each miRNA candidate and prioritizes those with stacking mature miRNA transcripts at both 5p and 3p arms of the precursor hairpin. The miRDeep2 [30] algorithm was run including miRNA homolog sequences annotated in human, bovine, and canine genome assemblies from the miRBase database v.22 [3] as a reference for comparison. This was motivated by their close evolutionary relationship to the feline miRNAome [7]. The resulting hairpins showing a miRDeep score ≥ 4 (representing an estimation of at least 21% true

positives) and stable secondary structure, represented by significant *p*-values after running the randfold algorithm [32], were kept as putative novel miRNA candidates.

To ensure that both known and putative novel miRNAs were not wrongly annotated resulting in potential false positives, they were subjected to further filtering based on sequence homology with other species as in Laganà *et al.* 2017 [7]. For this, we used the BLAST v.2.7.1 software [33] to query the feline miRNA mature sequence against the human, bovine, and canine mature sequences from miRBase [3], using "-perc_identity 70" (match over 70% of their nucleotide bases) and E-value < 0.1 as in Laganà *et al.* 2017 [7]. Both known and putative novel miRNAs were then annotated as feline orthologs of the miRNA precursors from the miRBase database [3] that retrieved the best hit.

## miRNA quantification and differential abundance analyses

miRNA quantification was assessed with the featureCounts v.1.5.3 tool [34]. Differential abundance analyses were carried out with the R package DESeq2 [35]. For analyses aside from differential abundance, normalization of expression values was performed with edgeR [36] using the trimmed mean of M values (TMM) [37] and then transformed into counts per million estimates (CPM). Two approaches were applied for differential abundance analyses: (i) Control vs. each of the urological pathologies and (ii) PN vs. each pathology individually and vs. all the other pathologies together (SB/C+UO+CKD). Correction for multiple testing of gene-wise *p*-values (*q*-values) after differential abundance analyses was performed with the false discovery rate (FDR) procedure reported by Benjamini and Hochberg [38]. Only differentially abundant (DA) miRNAs with an absolute value of fold change in the $\log_2$ scale ($|\log_2\text{FC}|$) $\geq 2$ and *q*-value < 0.05 were considered.

## Validation of differentially abundant (DA) miRNAs by qPCR

Small RNAseq results were cross-validated using high-throughput qPCR. In brief, cDNA was prepared in technical duplicates from each RNA sample previously used in the small RNAseq study. cDNA was synthesized according to Balcells *et al.* 2011 [39], using 15 ng RNA per reaction. Two noPAP controls (reactions carried out without poly(A) polymerase) using RNA from two different samples were likewise produced.

A panel of 96 different miRNAs was chosen for quantification using qPCR (S1 Table). miRNAs were primarily selected based on small RNAseq results, i.e. DA miRNAs (n = 59) and stably detected miRNAs (n = 11); those miRNAs that displayed the lowest coefficient of variation in small RNAseq data were included as potential endogenous miRNA normalizers (n = 10). A small subset was included due to being identified in relevant literature either as potential endogenous miRNA normalizers (n = 4) or highlighted as DA miRNAs in urological conditions (n = 8). Additionally, miRNAs experimentally validated to target neutrophil-recruiting chemokines (*CXCL2*, *CXCL8*, *CCL3*) were also included (N = 4) due to relevance in bacterial inflammation [40]. These miRNAs targeting relevant chemokines were identified using TarBase v.8 [41] and miRTarBase v.8.0 [42] databases. qPCR primers were designed for each miRNA using the miRprimer software [43]. cDNA samples were pre-amplified with 21 cycles prior to qPCR following manufacturer's instructions. Pre-amplified samples were then treated with exonuclease I (New England BioLabs) to digest any residual primers. qPCR was carried out on the high-throughput platform BioMark HD (Fluidigm) using an integrated fluidic circuit (IFC) 96.96 GT Dynamic Array chip and following manufacturer's recommendations. The 96.96 Dynamic Array contained all pre-amplified cDNA samples, including noPAP controls, a non-template control (NTC), and three independent dilution series made from a pool of all pre-amplified cDNA samples for qPCR efficiency estimation.

qPCR data was subsequently manually curated using the appertaining Real-Time PCR Analysis software (v.4.7.1, Fluidigm). Briefly, assays displaying unspecific amplification in their melting curves, assays showing very low abundance and high proportion of many missing values, assays with poor qPCR efficiency (outside the range 80–110%), or assays for which sample Cq overlapped with the Cq of the appertaining noPAP control were excluded from further analyses. The GenEx6 Pro software (MultiD Analyses AB) was used for qPCR efficiency correction, evaluation of potential endogenous miRNA normalizers, normalization to stable miRNA genes, averaging of cDNA replicates, and calculation of relative quantities and $\log_2$ transformation.

Principal component analysis (PCA) using $\log_2$ transformed relative quantities ($\log_2$Rq) for all successfully profiled miRNAs by qPCR was performed to detect possible clustering patterns according to disease condition and to the sex/neuter status.

$\log_2$Rq values for each profiled miRNA were further used for assessing differences in mean abundance across the defined comparisons, i.e. Control vs. PN, Control vs. SB/C, Control vs. UO, Control vs. CKD, PN vs. SB/C, PN vs. UO, PN vs. CKD, and PN vs. other pathologies (SB/C+UO+CKD). The comparison of PN vs. other pathologies was carried out using the Welch's t-test for unpaired groups of samples [44] implemented in the *t.test* R function [45]. Differences in mean abundance for the remaining comparisons were evaluated by implementing a one-way analysis of variance (ANOVA) over each of the considered groups (i.e. Control, PN, SB/C, UO, CKD) followed by *post hoc* analysis of difference using the Tukey-Kramer honest significant test method for unevenly sized groups [46] implemented in the *TukeyHSD* R function [45]. After differential abundance analyses of qPCR profiles, miRNAs with $|\log_2 FC|$ ≥ 1.5 and q-value < 0.05 were considered as showing significant differential abundance between groups.

Additionally, the agreement between abundance profiles of DA miRNAs according to qPCR analyses ($|\log_2 FC|$ ≥ 1.5; q-value < 0.05), as reported in Jessen *et al.* 2020 [17], and those obtained with small RNAseq ($|\log_2 FC|$ ≥ 2; q-value < 0.05), was assessed. To this end, we computed pairwise Pearson´s correlations and Bland-Altman plots between $\log_2$(Rq) values for qPCR and $\log_2$(CPM) values for small RNAseq data for each of the miRNAs that survived multiple testing and FC cut-offs in each respective comparison.

## Assessment of miRNA origin in urine by tissue clustering

We investigated the tissue origin from where quantified miRNAs in urine were secreted. For this purpose, we used a curated miRNA expression atlas from different tissues in the dog. We chose the dog over other more comprehensive tissue collections available, for instance from humans or mice, because the dog is phylogenetically closer to the cat. Therefore, we would expect a more conserved tissue-specific miRNA expression pattern between these two carnivore species, dogs and cats, compared with other less related species such as humans. Available miRNA profiles in cat tissues were scarce and not considered of high quality, so we favored the dog atlas for our analyses. Consequently, small RNAseq data from a tissue atlas in dog [47,48] was collected, and reads were quality-checked and trimmed using Cutadapt v.1.0 software [25]. The miRNA annotation in the dog species was retrieved from the miRGeneDB 2.1 website [4], and reads were mapped against mature miRNA sequences using the Bowtie v.1.2.3 aligner [26]. miRNA abundance was then determined for each considered tissue (brain, colon, duodenum, jejunum, ileum, plasma, heart, skeletal muscle, skin, kidney, liver, lung, and pancreas) with the featureCounts v.1.5.3 tool [34], and normalized by using z-scored CPM estimates. The uniform manifold approximation and projection for dimension reduction (UMAP) algorithm [49] was then implemented for sample clustering according to the miRNA

profiles in each canine tissue. Subsequently, the abundances of each quantified miRNA from feline urine samples were kept if shared with the canine miRNA complement according to miRGeneDB dog annotation [4], normalized as z-scored CPMs and projected onto the UMAP clusters based on canine tissues.

### Prediction of miRNA target genes and pathway enrichment analysis

After finding UMAP clustering of our feline urine samples with relevant canine tissues, we investigated miRNA-mRNA interactions for a selection of miRNAs identified as DA between PN and other urological conditions in both small RNAseq and qPCR analyses, in order to infer putative biological consequences of their altered abundance patterns. The TargetScan webserver software [50] was used to predict miRNA binding sites in the 3'UTR of mRNA targets using the human 3'UTRs as a reference. Target mRNAs showing the top 1,000 TargetScan context++ scores were considered. The context++ score by Agarwal *et al.* [50] includes information of 14 estimated features in order to rank the predicted target sites as biologically functional. Additionally, annotated 3'UTRs from the domestic cat genome assembly (Felis_catus_9.0) were retrieved from the bioMart database (www.ensembl.org/biomart/martview/) and used as reference for miRNA binding sites prediction. We applied the seedVicious 1.0 tool [51] to predict miRNA binding sites of type 8mer, 7mer-m8, and 7mer-A1 over the feline 3'UTRs. Predicted miRNA-mRNA interactions shared by both methods (TargetScan using human 3'UTRs and SeedVicious using feline 3'UTRs) were considered as reliable and kept for further analyses.

Subsequently, putative target mRNAs selected for each DA miRNA according to both qPCR and small RNAseq were used as inputs for pathway enrichment analysis. The ClueGO v.2.5.7 plug-in application [52] embedded in the Cytoscape v.3.6.0 software [53] was used to determine enriched pathways based on the Reactome and KEGG databases using the human annotation as reference. Enrichment significance was assessed with a right-sided hypergeometric test. Only enriched pathway terms with $q$-values $< 0.05$ after multiple testing correction with the FDR method [38] were considered significant.

## Results

### Cat population

Thirty-eight cats were included in the current study and were classified into five categories: healthy Control cats (n = 11), cats with PN (n = 10) and cats with other urological conditions (n = 17), consisting of cats with SB/C (n = 5), cats with UO (n = 7) and cats with CKD (n = 5). Of the 38 cats, 29 had participated in our previous study [17], and 9 were recruited specifically for the current study. Descriptive data from the cat population in the study is summarized in Table 1.

RNA extraction was successfully performed on all urine samples and RNA concentration ranged between 8.2–66.5 (ng/μ) (S2 Table).

### Small RNAseq data analysis

**Quality control of the small RNAseq.**  Thirty-eight feline urine samples were individually sequenced using small RNAseq. On average, 13.6M reads were obtained for each small RNAseq library. Roughly 93.3% of the reads passed quality control filters, and of these, 58.3% mapped to the feline genome. After removal of the reads mapping to the gene biotypes rRNA (average 0.5% of the mapped reads), tRNA (28.4%), snoRNA (1.5%), snRNA (1.0%) and RE

(27.4%) (S3 Table; S2 Fig), the remaining reads were used for known and novel miRNA identification.

**Known and de novo miRNA identification in urine.** In order to identify the miRNA transcripts present in feline urine, available annotation for known feline miRNAs and software for prediction of putative novel miRNAs were employed.

In the small RNAseq dataset, 146 previously annotated feline miRNAs were identified. Eighty-two corresponded to the Ensembl database annotation [54] and 64 to those detected by Laganà *et al.* 2017 [7]. After the *de novo* miRNA prediction, the miRDeep2 algorithm returned 996 putative novel miRNA candidates (S4 Table); from these, only 213 passed quality control filters (cut-off score $\geq 4$ and significant randfold *p*-values for miRNA hairpin folding stability).

We performed an additional *in silico* homology analysis using conserved miRNAs from humans, cattle, and dogs, which are annotated in miRBase [3]. For the known miRNAs, 137 out of the 146 (93.84%) miRNAs identified had a mature miRNA sequence highly conserved in the other species. More specifically, 104, 12, and 21 miRNAs showed complete homology with cattle, dog, and human miRNAs, respectively. For the putative novel miRNAs predicted, 75 out of 213 (35.21%) candidates displayed identical conserved mature sequences with 32, 5, and 38 candidate miRNAs from cattle, dogs and humans, respectively (S5 Table). The final number of miRNAs that passed the homology filter and were kept for further analysis were 212, 137 corresponding to previously known feline miRNAs and 75 novel putative novel miR-NAs with homology evidence from our *de novo* prediction analysis (S5 Table).

Principal component analysis (PCA) based on the 212 miRNA abundance profiles of each of the 38 urine samples showed that the first component explained 66% of the total variance, and a clear separation of 3 Control samples (C6, C7, C8) from the rest of the samples (S3A Fig). When including these 3 samples in the differential abundance analyses, we obtained 2- to 3-fold more DA miRNAs than when excluding them, pointing towards exceptional variation for these samples. It was considered that they might have suffered from an unnoticed technical problem during sample processing, leading to miRNA abundance profiles deviating from the other animals belonging to the Control group. Therefore, the 3 conflicting Control samples (C6, C7, and C8) were excluded from further analyses based on RNAseq data. For the rest of the samples, PCA showed weak evidence of sample clustering according to health condition (S3B Fig). Moreover, PCA of the effect of the sex/status on the RNAseq data did not show evidence of clustering.

Overall, the known miRNAs were identified in a higher proportion of the samples and with average higher abundance (S4A and S4B Fig) than the putative novel candidates (S4C and S4D Fig). The urinary miRNAome generally showed a low abundance profile. Of the 212 miRNAs queried, 159 showed $\geq 1$ CPM and only 63 miRNAs had $\geq 50$ CPM on average (S6 Table). Indeed, the top 15 most highly abundant miRNAs accounted for nearly 71% of all detected miRNA reads. These 15 miRNAs showed average abundances over 980 CPM each and were annotated predominantly based on the Ensembl and the feline miRNAome databases (Tables 2 and S6). The vast majority of these 15 miRNAs were also identified in kidney by Laganà *et al.* 2017 [7], including miR-320a, miR-99a, miR-21, miR-10b, miR-423, miR-378, let-7g, miR-200a, miR-19, and miR-200b-3p as highly abundant, and miR-146a and let-7i, which were moderately abundant.

**Differential abundance analysis by RNAseq.** Two different objectives were queried for the differential abundance analyses based on small RNAseq miRNA quantification:

i. Identification of DA miRNAs between healthy Control samples and each of the pathologies studied. With this first objective, a total of 41 DA miRNAs were identified in the Control vs.

**Table 2. List of the 15 most expressed miRNAs in feline urine using small RNAseq.**

| miRNA ID | Database/Study | Genomic Coordinates (Chr:start-end) | Mean (CPM) | SD (CPM) |
|---|---|---|---|---|
| miR-320a | Ensembl | B1:35931472–35931528 | 5,475.0 | 13,389.1 |
| miR-30a-5p | Ensembl | B2:67306469–67306533 | 4,355.0 | 2,811.7 |
| miR-146a-3p | Ensembl | A1:190133526–190133582 | 2,191.4 | 4,589.9 |
| miR-99a-5p | Laganà *et al.*, 2017 | C2:27213168–27213227 | 2,172.0 | 2,638.4 |
| miR-21-5p | Ensembl | E1:29105145–29105206 | 2,137.4 | 1,673.3 |
| miR-10b | Ensembl | C1:166919922–166919984 | 1,855.9 | 1,110.0 |
| miR-423-5p-2 | Laganà *et al.*, 2017 | E1:16813342–16813402 | 1,592.3 | 2,572.7 |
| miR-378c-1 | Ensembl | A1:199394444–199394500 | 1,452.7 | 2,802.8 |
| miR-30e-5p | Ensembl | C1:31853394–31853459 | 1,368.6 | 564.5 |
| let-7i | Ensembl | B4:90431559–90431638 | 1,256.7 | 1,476.6 |
| let-7g | Laganà *et al.*, 2017 | A2:20866213–20866293 | 1,083.6 | 1,019.2 |
| miR-200a | Laganà *et al.*, 2017 | C1:279939–280001 | 1,074.7 | 705.1 |
| miR-30d | Novel | F2:78073579–78073642 | 1,071.2 | 554.4 |
| miR-191b | Ensembl | A2:18165981–18166043 | 990.3 | 735.5 |
| miR-200b-3p | Ensembl | C1:279235–279295 | 981.3 | 544.7 |

CPM: Counts Per Million; SD: Standard Deviation.

PN analysis, 20 for Control vs. SB/C, 13 for Control vs. UO and 25 for Control vs. CKD (Fig 1A and S7A and S7B Table), respectively. In total, we found 71 DA miRNAs using this first approach with some of these miRNAs being identified in more than one comparison, as depicted in Fig 1A.

ii. Identification of DA miRNAs between PN cats and each of the other pathologies studied, as well as all the other pathologies grouped (SB/C+UO+CKD). In total, we found 59 DA miRNAs by this second approach. Seven DA miRNAs were found in the PN vs. SB/C comparison, 16 for PN vs. UO, 51 for PN vs. CKD, and 28 for PN vs. all the other pathologies (Fig 1B and S7A Table). Of these, the number of potential miRNA biomarkers specific for each condition were 2 (PN vs. SB/C), 4 (PN vs. UO), 18 (PN vs. CKD) and 1 (PN vs. other pathologies) (Fig 1B and S7A Table).

## Validation of differentially abundant miRNAs by qPCR

**Quality control of qPCR data.** Of the 96 initial miRNA assays profiled by qPCR, 57 were excluded from further analyses due to one or more of the following issues encountered upon initial inspection of the raw qPCR data: (i) unspecific amplification evident from melting curve; (ii) qPCR efficiency outside the accepted range (80–110%); (iii) very low abundance and/or excess of missing values (no Cq); or (iv) sample Cq value overlapping with Cq of noPAP control. Thirty-nine assays out of the 96 (40.62%) were accepted for further processing. Of these, three assays (miR-10a, miR-30a-5p and miR-200a-3p) were stably quantified across samples according to GeNorm [55] and NormFinder [56] algorithms and were thus selected as normalizers, leaving a total of 36 miRNAs to be analyzed by differential abundance analyses using qPCR data.

**Differential abundance analysis of qPCR validated miRNAs.** After performing differential abundance analyses on qPCR miRNA profiles, 9 miRNAs were detected as DA ($|\log_2 FC| \geq 1.5$; *q*-value $< 0.05$) in the Control vs. PN contrast. Of these, miR-125b-5p, miR-204, and miR-125a-5p were down-regulated, while miR-24-3p, miR-27a-3p, miR-21-5p, miR-27b, miR17-

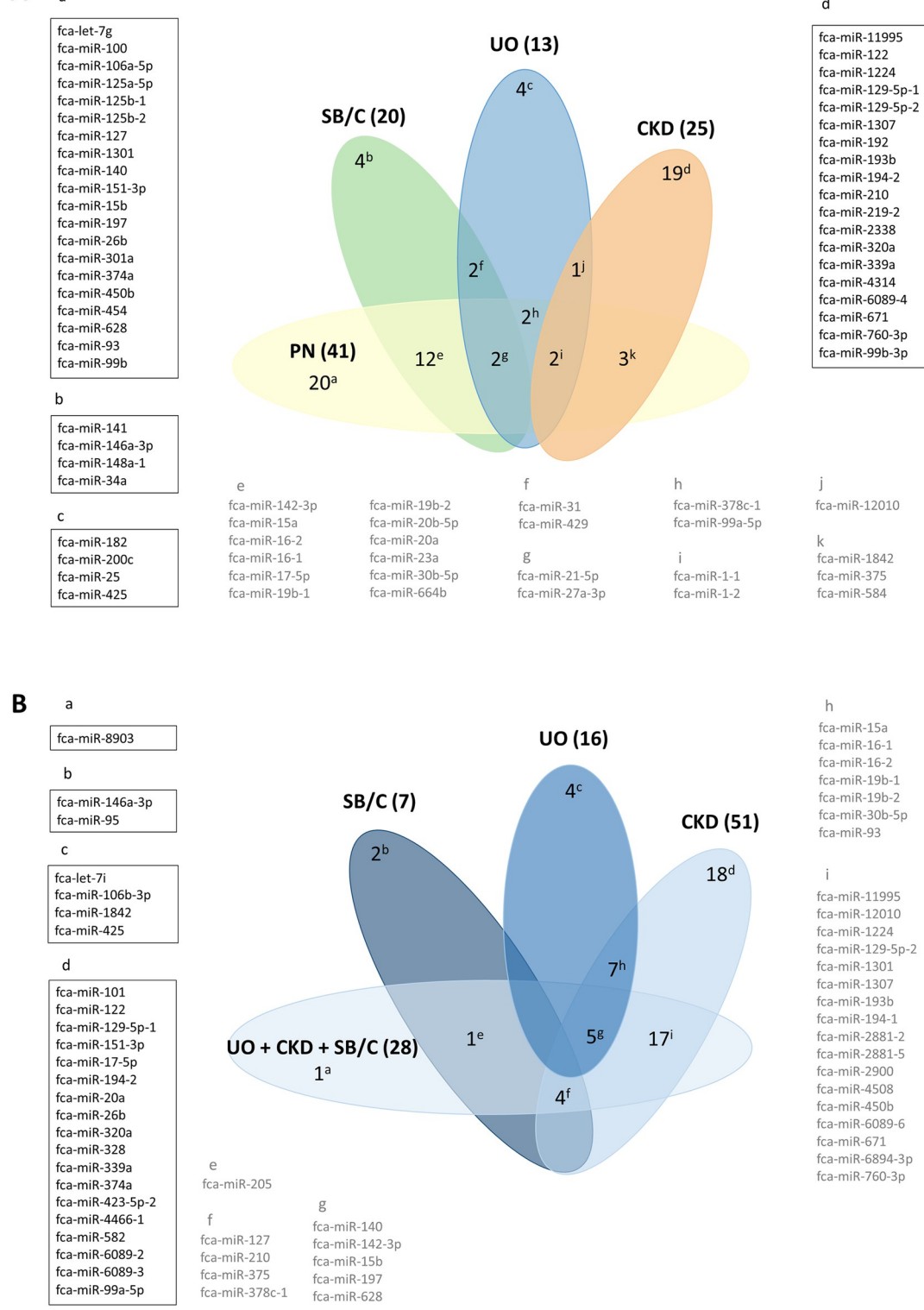

**Fig 1. Venn diagram of the differentially abundant (DA) miRNAs in each of the analyzed comparisons. A.** Shared and private miRNAs when comparing healthy cats (Control) with other pathologic urological conditions. In brackets, the total number of DA miRNAs in each comparison between healthy cats (Control) and cats with PN, SB/C, UO or CKD. **B.** Shared and private miRNAs when comparing cats with PN and cats with other urological conditions. In brackets, the total number of DA miRNAs in each comparison between PN cats and cats with SB/C, UO, CKD and all these disease states together (SB/C

+UO+CKD). The miRNAs found to be shared or private in each defined comparison are provided within each comparison with a superindex letter. miRNAs shown in a squared box and highlighted in black correspond to DA miRNAs specific for each group (private). In light grey, the DA miRNAs shared between 2 or more of the defined contrasts. CKD: Chronic Kidney Disease; PN: Pyelonephritis; SB/C: Subclinical Bacteriuria /Cystitis; UO: Ureteral Obstruction.

5p, and miR23a-3p were up-regulated in PN cats. When comparing Control vs. SB/C individuals, 3 miRNAs were highlighted: miR-204 and miR-125b-5p were down-regulated in SB/C, while miR-146b-5p was up-regulated in SB/C cats. (Table 3). The remaining comparisons did not result in any miRNAs being detected as DA. The full list of 36 miRNAs queried by qPCR after differential abundance analyses is available at S8 Table.

When not applying the FDR correction threshold for prioritizing DA miRNAs, several additional miRNAs were found at the nominal level of significance ($p$-value $< 0.05$) and with $|\log_2FC| \geq 1.5$, including 6 miRNAs relevant for discriminating PN from other urological conditions alone or in combination (miR-30c, miR-30b-5p, miR-30e-5p, miR-27a-3p, miR-27b-3p and miR-222) as shown in Table 3.

Fold change values in the $\log_2$ scale ($\log_2FC$) are computed using the Control cats as baseline, such as any positive $\log_2FC$ implies an up-regulation of the specific miRNA in the specific disease state and vice versa. Accordingly, in the comparison PN vs. other pathologies, the PN group was set as baseline, and a negative FC thus indicates up-regulation in PN compared to other pathologies. Only miRNAs with $|\log_2FC| \geq 1.5$ are shown. The full list of miRNAs (n = 36) after differential abundance analyses is shown in S8 Table. In bold, miRNAs with false discovery rate (FDR) $q$-value $< 0.05$; *: miRNAs also found as DA in the small RNAseq analysis. CKD: Chronic kidney disease, PN: Pyelonephritis, SB/C: Subclinical bacteriuria/Cystitis, UO: Ureteral obstruction, FDR: false discovery rate.

The inspection of box plots for visualizing the abundance profiles of each DA miRNA ($|\log_2FC| \geq 1.5$; $q$-value $< 0.05$) found in qPCR analyses (Fig 2), showed clearly differentiated and consistent patterns for the most highly DA miRNAs. This included miR-125b-5p, miR-27a-3p, miR-21-5p, miR-27b-3p, miR-204, miR-125a-5p, miR-24-3p, miR-17-5p, and miR-23-a-3p in the Control vs. PN comparison or miR-204, miR-125b-5p and miR-146b-5p in the Control vs. SB/C comparison (Table 1 and Fig 2). Cats belonging to the PN and SB/C groups had infections caused by *E. coli*, *Staphylococcus spp.*, or other pathogens, while Control, UO, and CKD cats showed no evidence of infection after urine culture (Fig 2). Six additional miR-NAs (miR-30c, miR-30b-5p, miR-30e-5p, miR-27a-3p, miR-27b-3p and miR-222) relevant for discriminating PN from the other urological conditions were identified by qPCR at the nominal level of significance ($p$-value $< 0.05$) (Table 1 and Fig 2).

PCA on qPCR data only showed evidence for clustering according to disease condition for Control vs PN and Control vs SB/C contrasts (S5 Fig), a result matching the obtained DA miR-NAs displayed in Table 3. Furthermore, when analyzing the sex/neuter status, PCAs did not revealed any clustering, in agreement with the same analysis performed on RNAseq data.

## Correlation analysis between small RNAseq and qPCR

Correlation between qPCR and small RNAseq data was tested only for those miRNAs that were commonly DA by both techniques (Table 3), using Pearson correlation (S6 Fig) and Bland-Altman agreement plots (S7 Fig). It is worth mentioning the case of miR-204, which showed a $q$-value $< 0.05$ in the small RNAseq data, but did not pass the fold-change filtering criteria of $|\log_2FC| \geq 2$ (S6 Fig and S7B Table); however, we considered it relevant to be included in the correlation analysis. For the Pearson correlation analysis, only four out of eight miRNAs analyzed (miR-23a-5p, miR-27a-3p, mir-125b-5p, and miR-204) showed high

**Table 3. Differentially abundant (DA) miRNAs from qPCR analyses.**

| Comparison | miRNA ID | log$_2$FC | *p*-value | *q*-value (FDR) |
|---|---|---|---|---|
| **Control vs. PN** | **miR-125b-5p**$^*$ | -2.1379 | 9.98E-06 | 3.60E-04 |
| | **miR-27a-3p**$^*$ | 3.7859 | 2.53E-04 | 4.56E-03 |
| | **miR-21-5p**$^*$ | 2.5948 | 6.85E-04 | 6.58E-03 |
| | **miR-27b-3p**$^*$ | 3.3647 | 7.31E-04 | 6.58E-03 |
| | **miR-204** | -1.704 | 1.47E-03 | 1.06E-02 |
| | **miR-125a-5p**$^*$ | -1.6419 | 2.78E-03 | 1.67E-02 |
| | **miR-24-3p** | 1.9085 | 5.49E-03 | 2.82E-02 |
| | **miR-17-5p**$^*$ | 1.9084 | 8.99E-03 | 3.85E-02 |
| | **miR-23a-3p**$^*$ | 2.6373 | 9.63E-03 | 3.85E-02 |
| | miR-30b-5p$^*$ | 1.5982 | 1.80E-02 | 6.49E-02 |
| | miR-29c-3p | 2.4951 | 2.38E-02 | 7.80E-02 |
| | miR-16-5p$^*$ | 1.6861 | 3.31E-02 | 9.93E-02 |
| | miR-146b-5p | 1.9629 | 4.76E-02 | 1.24E-01 |
| **Control vs. SB/C** | **miR-204** | -2.4206 | 2.36E-04 | 8.51E-03 |
| | **miR-125b-5p**$^*$ | -1.9142 | 1.13E-03 | 2.03E-02 |
| | **miR-146b-5p** | 2.6336 | 2.77E-03 | 3.32E-02 |
| | miR-146a-5p | 1.9695 | 6.70E-03 | 6.03E-02 |
| | miR-200c$^*$ | 2.0524 | 8.45E-03 | 6.09E-02 |
| | miR-125a-5p$^*$ | -1.6248 | 2.36E-02 | 1.41E-01 |
| | miR-21-5p$^*$ | 2.4488 | 3.19E-02 | 1.64E-01 |
| **Control vs. UO** | miR-30b-5p$^*$ | 1.6379 | 2.43E-02 | 4.39E-01 |
| | miR-21-5p$^*$ | 2.2174 | 2.86E-02 | 4.39E-01 |
| **Control vs. CKD** | miR-194$^*$ | -2.2337 | 1.76E-02 | 6.35E-01 |
| | miR-30e-5p | -1.842 | 4.47E-02 | 8.04E-01 |
| **PN vs. SB/C** | miR-30c | -1.6753 | 8.45E-03 | 1.89E-01 |
| | miR-30b-5p | -1.9969 | 1.05E-02 | 1.89E-01 |
| **PN vs. UO** | - | - | - | - |
| **PN vs. CKD** | miR-30c | -1.615 | 1.42E-02 | 4.00E-01 |
| | miR-30b-5p$^*$ | -1.8426 | 2.22E-02 | 4.00E-01 |
| | miR-30e-5p | -1.8428 | 4.92E-02 | 5.91E-01 |
| **PN vs. other urological conditions** | miR-30e-5p | -1.5611 | 5.77E-03 | 1.04E-01 |
| | miR-27a-3p$^*$ | -1.6682 | 3.69E-02 | 1.97E-01 |
| | miR-27b-3p | -1.6605 | 4.25E-02 | 1.97E-01 |
| | miR-222 | -1.762 | 4.92E-02 | 1.97E-01 |

correlation between both methods ($r > 0.7$, *p*-value $< 0.05$, S6 Fig). Also, Bland-Altman plots showed that, overall, qPCR and small RNAseq results were concordant, as the majority of samples for each of the defined groups were located inside the 95% confidence interval (S7 Fig).

## Tissue clustering and urinary miRNA origin determination

A total of 53 canine samples belonging to 13 different tissues and comprising 476 profiled miRNAs were used for UMAP clustering. Besides, 138 out of 212 (65.1%) feline miRNAs quantified in our small RNAseq data from urine samples were successfully assigned to canine miRNAs and considered for projection onto the UMAP clusters of dog tissues. As depicted in Fig 3, 25 out of 35 feline urine samples (71.4%) clustered close to canine kidney samples. This tissue might conform to the cellular origin of the secreted miRNAs sequenced in urine (upper urinary tract in the

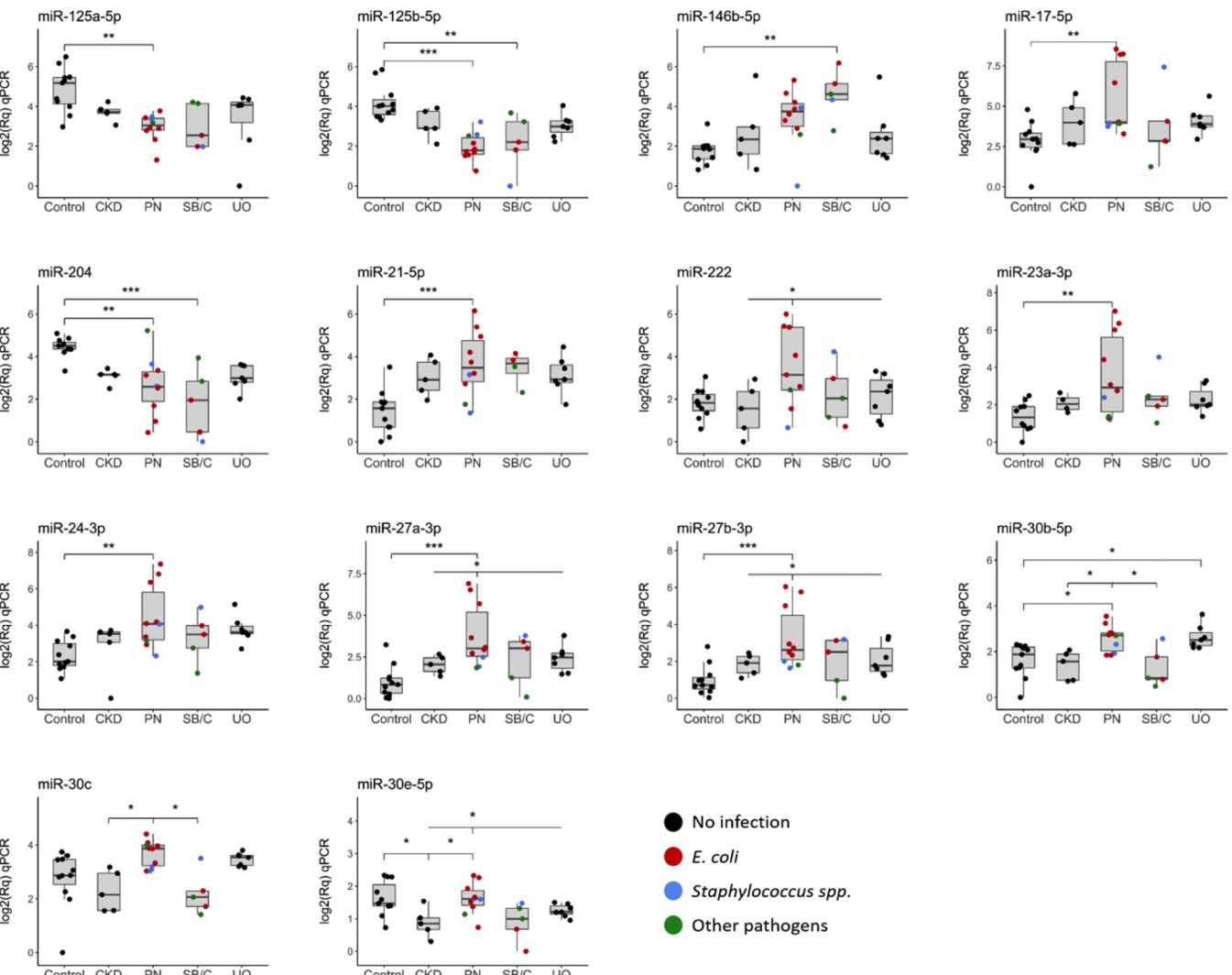

**Fig 2. Box-plot of differentially abundant (DA) miRNAs between healthy Control cats and each of the defined groups by using qPCR.** Selected miRNAs detected as DA ($|\log_2 \text{FC}| \geq 1.5$; $p$-value < 0.05) are shown. Cats with culture-positive urine after cystocentesis are highlighted in red (*E. coli*), blue (*Staphylococcus spp.*) and green (other pathogens), while cats with culture-negative urine are depicted in black. ***: DA miRNAs with $q$-value < 0.01. **: DA miRNAs with $q$-value > 0.01 & < 0.05. *: DA miRNAs with $p$-value < 0.05 but not significant after multiple testing correction ($q$-value > 0.05). PN: Pyelonephritis, SB/C: Subclinical bacteriuria/Cystitis, UO: Ureteral obstruction, CKD: Chronic kidney disease.

kidney). From the remaining samples that did not cluster close to kidney tissue, only five urine samples showed displaced clustering close to skin, and other additional five samples clustered close to other tissues, all of them not directly related, in principle, with the urinary tract.

## Pathway enrichment analyses

The DA miRNAs between Control and PN cats commonly found by small RNAseq and qPCR techniques were used for enrichment analysis (n = 7; miR-125a-5p, miR-125b-5p, miR-27a-3p, miR-27b-3p, miR-21-5p, miR-23a-3p, and miR-17-5p; Table 3, S7A and S8 Tables). As the miRNA pairs miR-125a-5p/miR-125b-5p and miR-27a-3p/miR-27b-3p shared their seed sequence, they were considered redundant. The non-redundant seeds of these shared 5 DA miR-NAs (miR-125a-5p, miR-27a-3p, miR-21-5, miR-23a-3p and miR-17-5p) were selected for

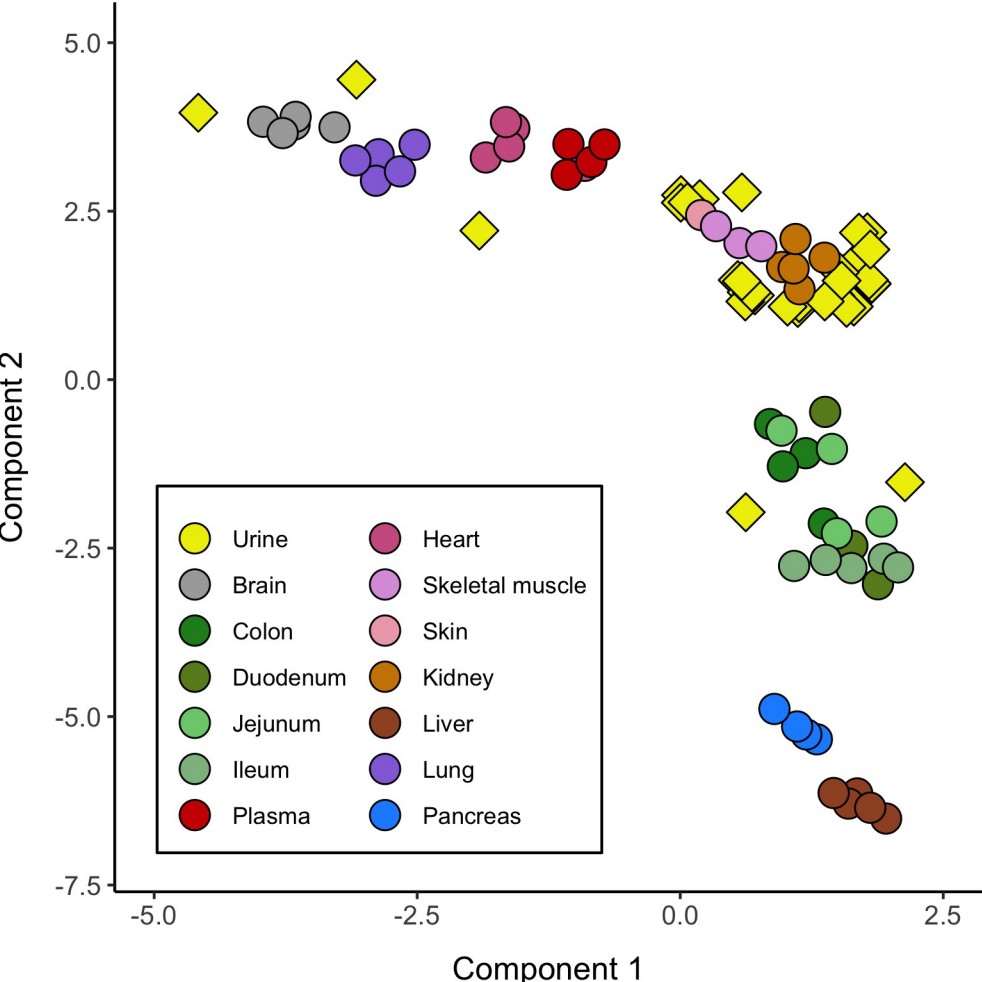

**Fig 3. UMAP plot depicting sample clustering of a collection of tissues from the canine miRNA atlas and the feline urinary samples analyzed in the current study.** The plot includes dog expression miRNA profiles from brain, colon, duodenum, jejunum, ileum, plasma, heart, skeletal muscle, skin, kidney, liver, lung and pancreas (round shape), as well as the predicted projection of the feline urinary samples using small RNAseq data (named Urine, diamond shape), including healthy cats (Control, n = 8) and cats with urological pathological conditions (PN, SB/C, UO or CKD, n = 27).

miRNA-mRNA interaction prediction using the TargetScan algorithm with human reference and search of miRNA binding sites in feline 3'UTRs. After target prediction and filtering of reliable putative mRNA targets for these miRNAs, pathway enrichment analyses were carried out to infer altered metabolic functions derived from abundance changes observed in miRNAs. Our results revealed the TGF-β signaling pathway as enriched in all putative mRNA targets from the miRNA seeds queried except for miR-125b-5p. It is also noteworthy that putative mRNA targets from miR-17-5p were related to bladder cancer, and both putative mRNA targets from miR-17-5p and miR-23a-3p were related to renal cell carcinoma (S9 Table).

## Discussion

In this study, we have assessed for the first time the feline urinary miRNAome using small RNAseq, and subsequent qPCR validation has allowed us to suggest a panel of miRNAs as potential non-invasive biomarkers associated with urological conditions in cats.

## The feline urine miRNAome

At present, none of the most widely used reference miRNA databases, i.e. miRBase [3], miR-GeneDB [4] and miRCarta [5] include feline miRNAs. Researchers thus have to use alternative annotation databases such as Ensembl (www.ensembl.org) or to use computational software to perform *de novo* prediction of feline miRNAs based on sequence information. This requires computational infrastructure and skilled bioinformatic work and brings the risk of falsely identifying novel feline miRNAs. In the present study we have applied a stringent approach by using the Ensembl database, the annotation from the study by Laganà *et al.* 2017 [7], and further filtering by homology analysis (BLAST) with well annotated genomes (human, cattle, and dog), in order to increase the set of known miRNAs in the domestic cat. This allowed a more comprehensive representation of the miRNA diversity present in urine, while controlling for the presence of false positive miRNA-like sequences that could bias the miRNA abundance profiling.

The analysis of small RNAseq data from cat urine revealed low abundance of urinary miRNAs compared to tRNA (S3 Table). These results are similar to previous findings in small RNAseq studies of human urine [57] or total RNAseq studies [58]. Indeed, the low proportion of reads assigned to miRNA genes in cat urine (1.7%; S3 Table) resembles similar findings in other species with high quality annotated genomes such as the human genome (3.2%) [57].

Our analysis of the small RNAseq data identified a total of 212 miRNAs present in cat urine. This number includes previously annotated and putative novel candidate miRNAs (S5 Table). Several of these miRNAs have also been described as highly abundant in feline urinary exosomes [16]. Interestingly, a considerable number of the novel candidate miRNAs were present at lower levels when compared to the already annotated miRNAs (S4 Fig). However, our *de novo* approach identified several miRNAs with high biomarker potential in urine and, therefore, we consider these results reliable; for example, urinary miR-30d, which has been associated with kidney injury and proposed as a biomarker in mice [59], and urinary miR-92a, previously found as differentially expressed between Control and CKD patients in humans [60] (S6 Table).

## The origin of miRNAs in feline urine

Urine is produced in nephrons by blood filtering. It is then collected into the renal pelvis and conducted to the bladder through the ureters to be finally excreted via the urethra. Hence, the miRNA transcripts detected in urine may have been transcribed and secreted from any cellular source along this route. It was, therefore, pertinent to investigate whether the identified urinary miRNA profile could be traced to their tissue of origin, or whether it just represented a rather random selection of quantified miRNAs. miRNAs could have been carried to the urine from the ultrafiltrate in the nephrons, or they might have derived from kidney cells and secreted into urine during its production. Another possibility is that the epithelial cover from the kidney pelvis, ureters and bladder might have also contributed to the miRNA abundance in urine. Of note, we were able to infer a tentative kidney origin for the miRNAs detected in urine based on tissue clustering using their relative abundances. The majority of the analyzed urine samples (71.4%) from cats clustered close to kidney samples from the canine miRNA atlas used as a reference, while none of them clustered to plasma samples [5] (Fig 3). Moreover, the most abundant miRNAs were also detected in feline kidney tissue [7]. This result is important because if miRNAs can be traced to their cellular origin, changes in their abundance in urine could be used as an indirect estimate of their function within their originating cells. Indeed, pathway enrichment analyses of predicted mRNA targets for the most highly DA miRNAs in urine showed several kidney and bladder related functions.

## Differential abundance analysis

Very little is known about the role of miRNAs in cats with PN or other urological conditions (SB/C, UO, and CKD). Therefore, the secondary aim of our study was to identify, using qPCR, a panel of DA urinary miRNAs in cats with PN in order to establish discriminatory abundance profiles for diagnostic use.

In a preliminary study, our group identified four deregulated miRNAs in urine (miR-16, miR-30a, miR-4286, and miR-204) associated with feline PN by using a qPCR approach and testing only 24 miRNAs [17]. In the current study, we have investigated for the first time the whole urine miRNAome using small RNAseq to identify miRNAs associated to urological conditions in cats. When comparing results from the two studies, miR-16 was confirmed to be upregulated in PN vs Control cats in this study too, but the *p*-value did not survive multiple testing due to the higher number of assays (8 assays in the pilot study vs. 39 in this study). Surprisingly, for miR-30a, which was differentially abundant (down-regulated in PN) in the pilot study, we found it stably expressed in the current study among the different groups, and was in fact used as a normalizer together with miR-10 and miR200a-3p. As for miR-4286, it was not detected in the small RNAseq experiment and was not investigated further. On the other hand, miR-204 was confirmed again to be downregulated in PN vs Control cats (*q*-value < 0.05).

**Differentiating cats with urological conditions from healthy cats.** qPCR profiling of the top candidates from the small RNAseq analysis confirmed a panel of several miRNAs that were DA between healthy Control cats and cats with urological pathologies. The majority of DA miRNAs were detected when comparing healthy cats and cats with PN or SB/C (Table 3). The panel included miR-16, a miRNA regulating TLR-mediated inflammation and cytokine expression [61,62], which was also deregulated in Control vs. PN in our previous study [17]. When applying correction for multiple testing, a total of 10 DA miRNAs were identified (Table 3), of which seven were DA in Control vs. PN and one was DA in Control vs SB/C (miR-146b-5p), while the remaining two (miR-125b-5p and miR-204) were DA in Control vs. both conditions (Table 3, Fig 2). Seven of the 10 miRNAs were considered more robust as they were identified as DA by both small RNAseq and qPCR analyses (Table 3), namely miR-125b-5p, miR-21-5p, miR-27b-3p, miR-125a-5p, miR-17-5p, miR-23a-3p, and miR-27a-3p, and the following discussion on differentiation between healthy cats and cats with PN pertains to these.

miR-17-5p, miR-21-5p, and miR-23a-3p have all previously been described in the literature related to kidney and/or bladder diseases. We found miR-17-5p to be up-regulated in PN cats. This miRNA targets genes involved in bladder cancer and renal cell carcinoma [63]. In the context of bacterial diseases, miR-17-5p has been shown to down-regulate lipopolysaccharide (LPS)-induced Toll-like-receptor 4 (TLR4)-mediated inflammation [64,65], thus favoring a balanced immune response. LPS is a powerful stimulator of inflammation derived from gram-negative bacteria including *E.coli*, the most prevalent pathogen causing urinary tract infections. In the urinary tract, TLR4 plays a central role in the innate host immune defense against *E.coli* [66], and its regulation is crucial to the clinical outcome of infection [67,68].

miR-21-5p was also up-regulated in PN cats compared with healthy Control cats. This miRNA is mostly expressed in the adult kidney [69] and is commonly deregulated in many kidney diseases (for a review see [70]), including kidney fibrosis and CKD. Accordingly, our results emphasized the broadness of miR-21-5p in urological conditions, as this miRNA was also up-regulated in the urine of cats suffering from UO and SB/C relative to the Control group.

miR-23a-3p was similarly present at increased levels in PN cats compared to healthy animals. The levels of miR-23a-3p have also previously been shown to be significantly increased in urinary exosomes and kidney tissue in human diabetic kidney disease, which is the most common cause of CKD in humans [71].

Finally, miR-27b-3p was also present at higher levels in PN cats when compared to healthy cats (Table 3). This miRNA has been found down-regulated in kidney biopsies and urine of human patients with diabetic nephropathy and also correlates with the degree of renal fibrosis [72]. It has been described in *M. tuberculosis* infections, in which it plays a dual role by increasing apoptosis and ROS production and blunting NF-kB activity [40].

miR-125b-5p and miR125a-5p, which share seed sequences, were found down-regulated in PN when compared to Control cats as well as in Control compared to SB/C cats. Both miRNAs have been proposed as important regulators for innate immunity and inflammation in different inflammatory diseases [73]. miR-125b-5p is a regulator of the NF-kB pathway, and its down-regulation may be a host strategy to prevent an excessive immune response to infection. It has also been proposed, however, as a pathogen strategy to survive in infected cells [40].

**Differentiating cats with pyelonephritis from cats with other urological conditions.**
We identified an additional set of six up-regulated miRNAs relevant for differentiating PN from other urological conditions based on qPCR analysis (miR-30b-5p, miR-30c, miR-30e-5p, miR-27a-3p, miR-27b-3p and miR-222), and, although they were only significant at the nominal level ($p$-value $< 0.05$), they deserve further discussion due to their potential as diagnostic discriminators in a clinical context.

miR-27a-3p, which shares the seed sequence and probably function with miR-27b-3p, was up-regulated in PN both when compared to Control cats and to cats with other urological conditions combined. It was the only DA miRNA ($p$-value $< 0.05$) between PN and other urological conditions combined to be identified both by small RNAseq and qPCR (Table 3 and S7B and S8 Tables), and it showed a highly correlated profiling ($r = 0.74$) between methods (S6 Fig). This miRNA is a known oncomiR that promotes cell proliferation and cycle progression [74]. Several studies have shown up-regulation of miR-27a-3p, e.g. in renal cell carcinoma and bladder cancer [75]. Moreover, miR-27a-3p was found to target genes involved in the TGF-β signaling pathway (S9 Table). This pathway plays a crucial role in inflammation-related tissues and is implicated in the regulation of cell proliferation, hypertrophy, apoptosis, and fibrogenesis [76]. Besides, miR-27a up-regulation has been associated with lowered inflammation and cell adhesion in acute renal injury by ischemia/reperfusion, via inhibition of TLR4 [77]. Negative regulation of TLR4 by miR-27a-3p has also been shown in Mycobacterial infections [40].

miR-222 was also up-regulated in PN versus other urological conditions combined. This miRNA has been shown to play a role in the link between *H. pylori* infection and gastric cancer. Being up-regulated in *H. pylori* infection, miR-222-promoted proliferation and invasion of gastric cancer cells [40]. Likewise, miR-222 promoted proliferation of epithelial cells when up-regulated by LPS in a gram-negative pneumonia model [78].

The miR-30 family was represented among miRNAs discriminating PN from its differential diagnoses when applying the criteria of $|log2FC| \geq 1.5$ and $p$-value $<0.05$ (Tables 3 and S8): miR-30b-5p and miR-30c were both up-regulated in PN compared to SB/C and CKD, while miR-30e-5p was up-regulated in PN both compared to CKD and compared to other urological conditions combined (Table 3). It is worth mentioning that these 3 miRNAs share the same seed sequence and, therefore, they probably exert the same function within cell metabolism. Similar to the other miRNAs identified in our study, the miR-30 family has not been reported in pyelonephritis. However, miR-30e was found to enhance innate immunity and reduce bacterial replication in an experimental model of uropathogenic *E.coli* [79]. Additionally, miR-

30c has been investigated in models of LPS-induced sepsis and its down-regulation is associated to kidney pyroptosis (inflammation-mediated apoptosis) [80], suggesting a potential role for this family in the regulation of the immune response to gram-negative bacterial injury as seen in PN.

Given the results found in this study, we were able to identify a potential set of miRNAs with differentiating profiles in the urine that could indicate the pathological health status of the cats. This panel was obtained by using whole-transcriptome sequencing of small RNA molecules followed by qPCR validation and increasing the number of analyzed cats compared to our previous pilot study [17]. With an extended sample size, we were able to refute and eliminate potential effects of sex and neuter status that were previously observed [17]. The proposed set of miRNAs could be easily applied to study larger cohorts of cats by standard RT-qPCR. This technique is a successful validation of more in depth techniques such as NGS, which is less cost-effective and maybe more suited for initial screening studies of fewer animals.

## Conclusion

We have characterized the urinary miRNAome in cats for the first time, identifying over 200 different miRNAs, including known and putative novel miRNAs. Several miRNAs were detected as DA when comparing healthy cats with cats suffering from urological conditions (PN, SB/C, UO, or CKD). Further qPCR verification allowed us to identify a panel of miRNAs that might allow discrimination between healthy cats and cats with these conditions. When examining differential abundance results between PN and other urological conditions (alone or in combination), an additional set of miRNAs that might serve to discriminate PN from its differential diagnoses were identified. However, only two of those were identified by both RNAseq and qPCR methods. The relatively few miRNAs identified to have discriminatory potential in our study may be a reflection of the multiple targets of individual miRNAs and the substantial overlap in the cellular pathways involved in the investigated pathologies. Thus, it might prove challenging to identify single urinary miRNAs with the ability to diagnose PN with high specificity, and a panel combining multiple markers (miRNAs and others) is likely necessary.

In summary, we identified a panel of DA miRNAs, including 7 discriminators between PN and healthy cats and 6 additional miRNAs with potential discriminating ability between PN and other urological conditions. Clinical validation of the identified urinary miRNA panel in an independent cohort is warranted, including evaluation of the application of the miRNA panel we described for monitoring disease regression/progression in cats with PN.

## Supporting information

**S1 Fig. Flowchart outlining the pipeline for small RNAseq analysis.** Including the identification of known and putative novel miRNAs, miRNA abundance profiling and differential abundance analysis. rRNA: ribosomal RNA; tRNA: transfer RNA; snoRNA: small nucleolar RNA; snRNA: small nuclear RNA; RE: repeat elements; qPCR: quantitative real-time PCR.
(TIF)

**S2 Fig. Stacked bar plot reporting the fraction of small RNAseq reads assigned to the annotated *Felis catus* miRNAs (fca-miRNAs) from Ensembl v.99 (blue), feline genome (orange) or that were not mapped (red).** CKD: Chronic kidney disease; PN: Pyelonephritis; SB/C: Subclinical bacteriuria/Cystitis; UO: Ureteral obstruction.
(TIF)

**S3 Fig. Principal component analysis (PCA) of samples profiled by small RNAseq technique. A.** PCA of urine samples on the basis of normalized read counts of the known and putative novel miRNAs for the 38 samples initially processed. The red arrows indicate the outlier Control samples (C5, C6 and C7). **B.** PCA excluding the high outlier samples. CKD: Chronic kidney disease; PN: Pyelonephritis; SB/C: Subclinical bacteriuria/Cystitis; UO: Ureteral obstruction.
(TIF)

**S4 Fig. Detailed characteristics of the known and putative novel miRNAs in cat urine for the 35 samples based on RNAseq data. A.** Proportion of samples for which each of the known miRNAs across the different groups were detected. **B.** Cumulative abundance of the known feline miRNAs. The dots indicate the log10 of the miRNA abundance for each miRNA. miRNAs are sorted in each group in a decreasing order by their miRNA abundance on the x-axis, independently for each group. **C.** Proportion of samples for which each of the putative novel miRNA candidates across the different groups were detected. **D.** Cumulative abundance of the putative novel miRNAs. The dots indicate the $\log_{10}$ of the miRNA abundance for each miRNA. miRNAs are sorted in each group in a decreasing order by their miRNA abundance on the x-axis, independently for each group. CKD: Chronic kidney disease; PN: Pyelonephritis; SB/C: Subclinical bacteriuria/Cystitis; UO: Ureteral obstruction, CPM: Counts per million.
(TIF)

**S5 Fig. Principal component analysis (PCA) of urine samples (N = 38) on the basis of $\log_2$ normalized relative quantities (Rq) of profiled miRNAs using qPCR.** All samples together (all groups), as well as each one of the contrasts considered (Controls vs. PN; Control vs. SB/C; Control vs. UO; Control vs. CKD; PN vs. SB/C; PN vs. UO; PN vs. CKD and PN vs. other Pathologies) are shown. CKD: Chronic kidney disease; PN: Pyelonephritis; SB/C: Subclinical bacteriuria/Cystitis; UO: Ureteral obstruction.
(TIF)

**S6 Fig. Pearson correlation analysis between abundance profiles of small RNAseq and qPCR data from selected miRNAs that were DA ($|\log_2 FC| \geq 1.5$ for qPCR and $\geq 2$ for small RNAseq; $q$-value $< 0.05$) using both methodologies.** CKD: Chronic kidney disease; PN: Pyelonephritis; SB/C: Subclinical bacteriuria/Cystitis; UO: Ureteral obstruction, CPM: Counts per million, Rq: Relative quantities.
(TIF)

**S7 Fig. Bland-Altman plots of abundance profiles of small RNAseq and qPCR.** The data presented is from selected miRNAs that were DA ($|\log_2 FC| \geq 1.5$ for qPCR and $|\log_2 FC| \geq 2$ for small RNAseq; $q$-value $< 0.05$) using both methodologies. CKD: Chronic kidney disease; PN: Pyelonephritis; SB/C: Subclinical bacteriuria/Cystitis; UO: Ureteral obstruction, CPM: Counts per million, Rq: Relative quantities.
(TIF)

**S1 Table. miRNAs selected for qPCR verification.** The table includes for each miRNA the arguments for its selection for further validation, the forward and reverse sequence, the miRBase sequence used as template for primer design, if successful miRNA amplification was obtained with qPCR and qPCR amplification efficiency.
(XLSX)

**S2 Table. Concentration of RNA obtained from whole urine.** Total RNA was isolated from whole urine of 38 cats, and 6 µl of each RNA sample was applied in small RNAseq.

Concentration and ratios were obtained using Nanodrop measurement.
(XLSX)

**S3 Table. Sequencing and mapping statistics.** Reads mapped to the different gene biotypes in the genome are included. SD: standard deviation; rRNA: ribosomal RNA; tRNA: transfer RNA; snoRNA: small nucleolar RNA; snRNA: small nuclear RNA; RE: repeat elements; miRNA: microRNA.
(XLSX)

**S4 Table. miRDeep2 results.** mirDeep2 output of the 996 miRNAs identified with information from their sequences, scores and probabilities of being a true positive miRNA, among others.
(XLSX)

**S5 Table. Detailed list of known and putative *de novo* miRNAs that passed the defined quality control filters in the feline urine datasets.** Including the miRNA ID name based on sequence homology with other species (blastn), miRNA coordinates, strand, closest homology miRNA with their percentage of identity and probability (E-value) and consensus mature, star and precursor sequences.
(XLSX)

**S6 Table. Mean abundance and standard deviation of the miRNAs detected in the urinary miRNAome of the final 35 feline samples included in RNAseq analyses.** This includes information from their source (database/study/de novo) and their genomic coordinates. CPM: Counts Per Million.
(XLSX)

**S7 Table.** A. List of differentially abundant (DA) miRNAs between the Control group (healthy) and the different uropathological conditions and between PN and the other uropathological conditions using RNAseq data. B. List of all miRNAs comparisons (without *q*-value and log2FC filters). The values of log2FC are calculated using the Controls as baseline and therefore a positive log2FC implies an upregulation of the specific miRNA in the pathological state and vice versa. Accordingly, in the comparison PN vs Other pathologies, the PN group was set as baseline, and a negative log2FC thus indicates up-regulation in PN compared to other pathologies. log2FC: log2 of the fold-change in expression. CKD: Chronic Kidney Disease; SB/C: Subclinical Bacteriuria/Cystitis; PN: Pylonephritis; UO: Ureteral Obstruction.
(XLSX)

**S8 Table. List of differentially abundant (DA) miRNAs between the healthy control group and the different uropathological conditions and between PN and the other uropathological conditions using qPCR data.** The values of $\log_2$FC are calculated using the Control cats as baseline, and, therefore, a positive $\log_2$FC implies an up-regulation of the specific miRNA in the pathological state and vice versa. Accordingly, in the comparison PN vs. other pathologies, the PN group was set as baseline, and a negative $\log_2$FC thus indicates up-regulation in PN compared to other pathologies. In dark grey DA miRNA surviving the criteria $|\log_2$FC$| \geq 1.5$ and *q*-value $< 0.05$; in light grey DA miRNAs surviving the criteria $|\log_2$FC$| \geq 1.5$ and *p*-value $< 0.05$; CKD: Chronic kidney disease; SB/C: Subclinical bacteriuria/Cystitis; PN: Pyelonephritis; UO: Ureteral obstruction, FDR: False discovery rate.
(XLSX)

**S9 Table. Pathway enrichment analysis of putative mRNA target genes of non-redundant differentially abundant (DA) miRNAs seeds surviving the criteria $|\log_2$FC$| \geq 1.5$ and *q*-**

**value < 0.05 in qPCR analyses and which were also DA in the small RNAseq dataset.** Pathway enrichment of the KEGG and Reactome terms are shown if they gathered significant false discovery rate (FDR) and corrected *p*-values (*q*-value < 0.05); their associated mRNA genes found are also shown.
(XLSX)

## Acknowledgments

The authors thank Tina N. Mahler for RNA isolation of the urine samples and the cat owners for allowing their cats to participate in this study.

## Author Contributions

**Conceptualization:** Rebecca Langhorn, Lise Nikolic Nielsen, Lisbeth Rem Jessen, Susanna Cirera.

**Data curation:** Marta Gòdia, Emilio Mármol-Sánchez, Susanna Cirera.

**Formal analysis:** Marta Gòdia, Emilio Mármol-Sánchez.

**Funding acquisition:** Bert Jan Reezigt, Lise Nikolic Nielsen, Lisbeth Rem Jessen, Susanna Cirera.

**Investigation:** Marta Gòdia, Louise Brogaard, Emilio Mármol-Sánchez, Rebecca Langhorn, Ida Nordang Kieler, Bert Jan Reezigt, Lise Nikolic Nielsen, Lisbeth Rem Jessen, Susanna Cirera.

**Methodology:** Louise Brogaard, Rebecca Langhorn, Ida Nordang Kieler, Bert Jan Reezigt, Lisbeth Rem Jessen.

**Project administration:** Lisbeth Rem Jessen, Susanna Cirera.

**Visualization:** Marta Gòdia, Emilio Mármol-Sánchez.

**Writing – original draft:** Marta Gòdia, Louise Brogaard, Emilio Mármol-Sánchez, Rebecca Langhorn, Lise Nikolic Nielsen, Lisbeth Rem Jessen, Susanna Cirera.

**Writing – review & editing:** Marta Gòdia, Louise Brogaard, Emilio Mármol-Sánchez, Rebecca Langhorn, Ida Nordang Kieler, Bert Jan Reezigt, Lise Nikolic Nielsen, Lisbeth Rem Jessen, Susanna Cirera.

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
