## [Decision Letter · Decision Letter 0]

2 Mar 2022

PONE-D-22-02085Urinary microRNAome in healthy cats and cats with pyelonephritis or other urological conditionsPLOS ONE

Dear Dr. Cirera,

Thank you for submitting your manuscript to PLOS ONE. After careful consideration, we feel that it has merit but does not fully meet PLOS ONE’s publication criteria as it currently stands. Therefore, we invite you to submit a revised version of the manuscript that addresses the points raised during the review process.

We look forward to receiving your revised manuscript.

Kind regards,

Silvia Sabattini

Academic Editor

PLOS ONE

Journal Requirements:

Reviewers' comments:

Reviewer's Responses to Questions

**Comments to the Author**

1. Is the manuscript technically sound, and do the data support the conclusions?

Reviewer #1: Yes

Reviewer #2: Yes

2. Has the statistical analysis been performed appropriately and rigorously? 

Reviewer #1: Yes

Reviewer #2: Yes

3. Have the authors made all data underlying the findings in their manuscript fully available?

Reviewer #1: Yes

Reviewer #2: No

4. Is the manuscript presented in an intelligible fashion and written in standard English?

Reviewer #1: Yes

Reviewer #2: Yes

5. Review Comments to the Author

Reviewer #1: The authors present a very thorough analysis of the miRnome of feline urine. The subject is of interest for veterinary dignostics and the knowledge of miRNAs as a diagnostic tool in cats is quite sparse, so any new insights in this field are advantageous. The methods and the results are presented clearly, and a lot of diligence has gone into the difficult questions of normalization of results and which miRNAs should be considered as biomarkers. However, in the discussion the authors focus exclusively on similarities in miRNA expression between the current study and previous studies. Since they take their own study from 2020 as a reference (number [17]), the authors should also mention the discrepancies. They should give an explanation why miR-30a is presented downregulated in the study of 2020, but considered a normalization miRNA in 2021 on the same (?) samples. Why was mir-4286 (from the 2020 study) not part of the panel? In the 2020 study, the authors also detected differences of sex and neuter status. This is not mentioned in the current study, although the authors speculated in 2020 on 'female/female neutered matched control groups' as 'directions for future investigations'. The manuscript also lacks a critical view on the discrepancies between the methods of RNAseq and quantitative PCR of miRNAs, e.g. mir-204 is missing in RNAseq, but prominent in PCR, is there an explanation? Which method would the authors recommend for larger studies, if discrimination of PN from other conditions via miRNA panel would be the diagnostic aim?

The discussion should be expanded to include these points.

Reviewer #2: Review, Research Article PONE-D-22-02085: “Urinary microRNAome in healthy cats and cats with pyelonephritis or other urological conditions”. The manuscript used small RNA sequencing to identify microRNA profiles for different urinary diseases in cats. The manuscript is, first, innovative, as this data is not currently available in the literature, and second, of great relevance to the field of veterinary medicine as cats frequently present with urinary diseases. The manuscript is very well written, and the flow and amount of data are adequate. The minimum sample per group is five, adequate for next-generation sequencing work in the discovery phase. The workflow is clearly stated and concise, and the results of this manuscript will be of great value in searching for more accurate non-invasive diagnostic tests (especially if early detected) in cats with urinary conditions.

There are a few questions/suggestions that should be addressed before publication, described below:

1: Please state if whole-urine was used instead of supernatant or sediment; I assume this was the case for this study.

2: If whole urines were used, there would be a significant difference in cellularity in the urine content depending on the condition. Is there a correlation regarding the presence of epithelial cells, leukocytes, and erythrocytes? Is urinalysis available?

3: The authors state that urine was collected from the bladder by cystocentesis and/or pyelocentesis. Could the different sample locations be a confounding factor? Perhaps the presence or absence of bladder cells could interfere with the differential expression as the samples are not comparable.

4: Please, add the concentration and quality of nucleic acid extracted from the urine; this is a common caveat of working with microRNAs in urine, and it would benefit the readers and reproducibility of the work.

5: The authors used NanoDrop to quantify the total RNA, which is not the most appropriate method; however, prior to library construction, it is likely that other methods were used, such as Bioanalyzer). If this is the case, please add these data to the manuscript (perhaps another Supplementary table).

6: The authors mentioned that 6 uL of the 30 uL elution was sent or the Genomics Unity; however, there is no information on the concentrations. The amount of miRNA extracted from control animals is likely much lower than samples with high cellularity, such as in inflammatory conditions, and this can be possibly correlated with technical errors. Generally, a minimum amount/concentration of RNA is required to proceed with sequencing. What was the concentration used?

6. PLOS authors have the option to publish the peer review history of their article (what does this mean?). If published, this will include your full peer review and any attached files.

Reviewer #1: No

Reviewer #2: No

---

## [Author Response · Author response to Decision Letter 0]

1 Apr 2022

Dear Academic editor Silvia Sabattini, 1st April, 2022

The authors thank the reviewers and the editor for their effort in reviewing this manuscript and for their valuable comments. We have addressed all the comments and below you will find our detailed response to the general and individual comments in red. Lines for the changes are provided according to the clean word document submitted.

Sincerely,

Susanna Cirera

Journal Requirements:

We confirm that this manuscript compiles with PLOS ONE’s style requirements and we have checked and updated the format where needed.

2.We note that you have stated that you will provide repository information for your data at acceptance. Should your manuscript be accepted for publication, we will hold it until you provide the relevant accession numbers or DOIs necessary to access your data. If you wish to make changes to your Data Availability statement, please describe these changes in your cover letter and we will update your Data Availability statement to reflect the information you provide.

We provide DOIs for all our data. See lines 760-761

“Data availability: The datasets generated and analyzed in the current study are available at NCBI’s BioProject PRJNA788683”. 

The data is currently sequestered and will be released immediately by the authors upon receiving notification of the manuscript being accepted for publication. 

3.PLOS requires an ORCID iD for the corresponding author in Editorial Manager on papers submitted after December 6th, 2016. Please ensure that you have an ORCID iD and that it is validated in Editorial Manager. To do this, go to ‘Update my Information’ (in the upper left-hand corner of the main menu), and click on the Fetch/Validate link next to the ORCID field. This will take you to the ORCID site and allow you to create a new iD or authenticate a pre-existing iD in Editorial Manager. Please see the following video for instructions on linking an ORCID iD to your Editorial Manager account: https://www.youtube.com/watch?v=_xcclfuvtxQ

Susanna Cirera, as corresponding author, has provided her ORCID iD (0000-0001-8105-1579) and it has been validated in Editorial Manager.

4.We note that you have included the phrase “data not shown” in your manuscript. Unfortunately, this does not meet our data sharing requirements. PLOS does not permit references to inaccessible data. We require that authors provide all relevant data within the paper, Supporting Information files, or in an acceptable, public repository. Please add a citation to support this phrase or upload the data that corresponds with these findings to a stable repository (such as Figshare or Dryad) and provide and URLs, DOIs, or accession numbers that may be used to access these data. Or, if the data are not a core part of the research being presented in your study, we ask that you remove the phrase that refers to these data

We have eliminated “data not shown” from line 397 because those results are not relevant for the results nor the discussion of the study.

We have also eliminated “data not shown” from line 520 and have included the data on an extra Table sheet in supplementary Table S7 (Table S7B).

5.Your ethics statement should only appear in the Methods section of your manuscript. If your ethics statement is written in any section besides the Methods, please move it to the Methods section and delete it from any other section. Please ensure that your ethics statement is included in your manuscript, as the ethics statement entered into the online submission form will not be published alongside your manuscript.

We have moved the Ethics statements to the methods in lines 121-123.

We have checked that the reference list is complete and with the correct format.

The references that contained the Issue number have been updated. No Issue numbers are included as required by the guidelines.

Reference (#4) has an updated page number and volume: 

4.Fromm B, Høye E, Domanska D, Zhong X, Aparicio-Puerta E, Ovchinnikov V, et al. MirGeneDB 2.1: toward a complete sampling of all major animal phyla. Nucleic Acids Res. 2021;gkab1101. Now is:

4.Fromm B, Høye E, Domanska D, Zhong X, Aparicio-Puerta E, Ovchinnikov V, et al. MirGeneDB 2.1: toward a complete sampling of all major animal phyla. Nucleic Acids Res. 2021;50: D204-D210

Reference (#70) now includes the page number:

70.Peters LJF, Floege J, Biessen EAL, Jankowski J, van der Vorst EPC. MicroRNAs in Chronic Kidney Disease: Four Candidates for Clinical Application. Int J Mol Sci. 2020;21: 6547.

Reviewers' comments:

Reviewer's Responses to Questions

Comments to the Author

1. Is the manuscript technically sound, and do the data support the conclusions?

Reviewer #1: Yes

Reviewer #2: Yes

2. Has the statistical analysis been performed appropriately and rigorously? 

Reviewer #1: Yes

Reviewer #2: Yes

3. Have the authors made all data underlying the findings in their manuscript fully available?

Reviewer #1: Yes

Reviewer #2: No

We are providing the DOI of all data and we have removed the statement “data not shown” and included the data where needed.

4. Is the manuscript presented in an intelligible fashion and written in standard English?

PLOS ONE does not copy edit accepted manuscripts, so the language in submitted articles must be clear, correct, and unambiguous. Any typographical or grammatical errors should be corrected at revision, so please note any specific errors here.

Reviewer #1: Yes

Reviewer #2: Yes

5. Review Comments to the Author

We would like to thank Reviewer #1 and Reviewer #2 for their time and effort in reviewing this manuscript and helping us to improve the quality of our study. We hope we have addressed all the concerns. 

Reviewer #1: 

-The authors present a very thorough analysis of the miRnome of feline urine. The subject is of interest for veterinary diagnostics and the knowledge of miRNAs as a diagnostic tool in cats is quite sparse, so any new insights in this field are advantageous. The methods and the results are presented clearly, and a lot of diligence has gone into the difficult questions of normalization of results and which miRNAs should be considered as biomarkers. However, in the discussion the authors focus exclusively on similarities in miRNA expression between the current study and previous studies. Since they take their own study from 2020 as a reference (number [17]), the authors should also mention the discrepancies. They should give an explanation why miR-30a is presented downregulated in the study of 2020, but considered a normalization miRNA in 2021 on the same (?) samples. Why was mir-4286 (from the 2020 study) not part of the panel? 

We agree that both similarities and discrepancies between the two studies should be included. Accordingly, we have added text commenting on that. See lines 627-635 in discussion section:

“When comparing results from the two studies, miR-16 was confirmed to be upregulated in PN vs Control cats in this study too, but the p-value did not survive multiple testing due to the higher number of assays (8 assays in the pilot study vs. 39 in this study). Surprisingly, for miR-30a, which was differentially abundant (down-regulated in PN) in the pilot study, we found it stably expressed in the current study among the different groups and it was, in fact, used as a normalizer together with miR-10 and miR200a-3p. As for miR-4286, it was not detected in the small RNAseq experiment and was not investigated further. On the other hand, miR-204 was confirmed again to be downregulated in PN vs Control cats (q-value < 0.05).” 

-In the 2020 study, the authors also detected differences of sex and neuter status. This is not mentioned in the current study, although the authors speculated in 2020 on 'female/female neutered matched control groups' as 'directions for future investigations'. 

Thank you for this comment. This current study is a continuation of the 2020 study. We therefore have the same cats included as well as the addition of nine new cats. It is true that we speculated on female/female neutered matched control groups in the 2020 study. However, with the current data set, we have performed Principal component analyses (PCA) on small RNAseq and qPCR data but it did not reveal any grouping in relation to sex and to neuter status on this larger study sample.

PCA of the effect of gender/status on the RNAseq data (figure provided in Response to reviewers document); No evident clustering by sex or neutering is observed.

PCA of the gender/status on RT-qPCR data results (figure provided in Response to reviewers document). Again, no evident clustering is observed.

We have now added a comment in the results (lines 512-514) and in the discussion (lines 728-730) about the sex/neuter status issue but we don’t think that is necessary to include the PCAs in supplementary files as no grouping is observed. 

-The manuscript also lacks a critical view on the discrepancies between the methods of RNAseq and quantitative PCR of miRNAs, e.g. mir-204 is missing in RNAseq, but prominent in PCR, is there an explanation? Which method would the authors recommend for larger studies, if discrimination of PN from other conditions via miRNA panel would be the diagnostic aim? The discussion should be expanded to include these points

The reviewer is right, but discrepancies mostly arise because different number of miRNAs were queried and this influenced the later statistical tests thresholds. For example, miR-204, presented a q-value = 0.009 and log2FC = 1.83 in DA analyses of small RNA-seq data (see newly added sheet B to Table S7). In this case, we did not include this miRNA in the reported results for RNAseq data as it did not reach the established threshold for minimum expression fold change observed (log2FC ≥ 2). We describe such result in lines 516-526 in the manuscript. There, we mention that we decided to include miR-204 in the correlation analyses between qPCR and small RNAseq data given the highly significant results obtained for this particular miRNA in qPCR and RNAseq analyses, despite the fact that miR-204 was not included itself in results reported originally. In order to clarify these results, we have now included a full list of DA miRNA analyses for RNA seq data in Table S7 (annexed sheet B).

Besides, we now provide in the discussion section further insights on which approach to take when trying to infer discriminant markers in larger studies. We have included the following paragraph in lines 724-733 to acknowledge this specific issue:

“Given the results found in this study, we were able to identify a potential set of miRNAs with differentiating profiles in the urine that could indicate the pathological health status of the cats. This panel was obtained by using whole-transcriptome sequencing of small RNA molecules followed by qPCR validation and increasing the number of analyzed cats compared to previous pilot studies [17]. With an extended sample size, we were able to refute and eliminate potential effects of sex and neuter status that were previously observed [17]. The proposed set of miRNAs could be easily applied to study larger cohorts of cats by standard RT-qPCR. This technique is a successful validation of more in depth techniques such as NGS, which is less cost-effective and maybe more suited for initial screening studies of fewer animals. 

Reviewer #2: 

Review, Research Article PONE-D-22-02085: “Urinary microRNAome in healthy cats and cats with pyelonephritis or other urological conditions”. The manuscript used small RNA sequencing to identify microRNA profiles for different urinary diseases in cats. The manuscript is, first, innovative, as this data is not currently available in the literature, and second, of great relevance to the field of veterinary medicine as cats frequently present with urinary diseases. The manuscript is very well written, and the flow and amount of data are adequate. The minimum sample per group is five, adequate for next-generation sequencing work in the discovery phase. The workflow is clearly stated and concise, and the results of this manuscript will be of great value in searching for more accurate non-invasive diagnostic tests (especially if early detected) in cats with urinary conditions.

There are a few questions/suggestions that should be addressed before publication, described below:

1: Please state if whole-urine was used instead of supernatant or sediment; I assume this was the case for this study.

Yes, whole urine was used. We have clarified this further in the updated version. See lines 157-158.

“Whole urine samples for urinalysis and for miRNA sequencing were collected from the bladder by cystocentesis”

2: If whole urines were used, there would be a significant difference in cellularity in the urine content depending on the condition. Is there a correlation regarding the presence of epithelial cells, leukocytes, and erythrocytes? Is urinalysis available?

As the reviewer correctly states, some cats, but not all, had an active sediment in the urinalysis, and depending on the underlying condition. In particular, cats with pyelonephritis, cystitis or subclinical bacteriuria have a very similar active sediment and this test can therefore not be used to differentiate these conditions. This was one of the reasons why we sought to find new biomarkers (miRNA) to improve this differentiation. The sediment analysis was performed at both veterinary facilities and consisted of cytological assessment of the cellular content as it is routinely performed in clinical practice. As authors, we are a bit unsure how a correlation between miRNAs and qualitative measures of cellular content would improve the study as the reviewer suggests. Firstly, we already know that the cellular content in infectious diseases cannot localize the condition to either the upper or lower urological tract. Secondly, we are aware that some of the miRNAs could be related to this cellular content, but with only a qualitative measure of the cellular content we are worried that any correlations would be over-conclusive.

3: The authors state that urine was collected from the bladder by cystocentesis and/or pyelocentesis. Could the different sample locations be a confounding factor? Perhaps the presence or absence of bladder cells could interfere with the differential expression as the samples are not comparable.

In cats with ultrasonographic evidence of a dilated renal pelvis, urine was obtained from the renal pelvis and used for bacterial culture and susceptibility testing to diagnostically rule in /rule out pyelonephritis. Urine for the miRNA analyses were all obtained from the bladder by cystocentesis, making the samples comparable. This is also stated in the Material and Methods paragraph called “Sampling, processing and storage of urine” lines 156-172.

4: Please, add the concentration and quality of nucleic acid extracted from the urine; this is a common caveat of working with microRNAs in urine, and it would benefit the readers and reproducibility of the work. 

We have now included an additional Supplementary Table (named S2 Table) with the concentration and nucleic acid quality ratios for each of the samples measured by Nanodrop. This has also been included in the text results section as lines 361-362: 

“RNA extraction was successfully performed on all urine samples, and RNA concentration ranged between 8.2 - 66.5 (ng/µ) (S2 Table)”.

Because an additional Supplementary Table has been created, succeeding table numbers have been updated across the manuscript.

5: The authors used NanoDrop to quantify the total RNA, which is not the most appropriate method; however, prior to library construction, it is likely that other methods were used, such as Bioanalyzer). If this is the case, please add these data to the manuscript (perhaps another Supplementary table). 

We completely agree with the reviewer, NanoDrop is not the best method for RNA quantification. Nevertheless, the RNA profile in urine samples has been shown to result in very low RNA Integrity (RIN) values in several studies. For example, “Bradley MS, et al., Female Pelvic Med Reconstr Surg. 2019;25(3):247-251”, showed duplicate urine samples showing RIN profiles of 3.3 and 5.9. And “Zhou K, et al., Anal Biochem. 2017;536:8-15”, showed the profile of 6 urine samples, with RIN values ranging between 1 and 2.6. Therefore, RIN values might not be indicative of the success of further sequencing by small RNAseq (RIN values are based on the 18S and 28S rRNA molecules and do not take into account the miRNA content). Given this situation, the sequencing unit suggested not performing further QC on the RNA samples as sampling material would be lost and it was important to have leftover RNA for RT-qPCR validation. Thus, unfortunately we cannot provide this information.

6: The authors mentioned that 6 uL of the 30 uL elution was sent or the Genomics Unity; however, here is no information on the concentrations. The amount of miRNA extracted from control animals is likely much lower than samples with high cellularity, such as in inflammatory conditions, and this can be possibly correlated with technical errors. Generally, a minimum amount/concentration of RNA is required to proceed with sequencing. What was the concentration used?

The average concentration between the different groups did not differ significantly: Control 18 ng/µl; CKD 11 ng/µl; PN 29 ng/µl; UO 11 ng/µl and SB/C 24 ng/µl. We do also believe that given the low concentrations, NanoDrop might be under/over estimating the concentrations and thus show some outliers.

After library prep with NEBNext® Small RNA Library Prep kit (New England Biolabs) samples are loaded in the sequencing unit after pooling with the exact concentration, and this quantification is done in triplicates with RT-qPCR. This means that even if one of the samples presented higher RNA concentration resulting in higher concentration after library prep, it would be brought to the same concentration levels as the others. Moreover, the concentration to be loaded in the Illumina HiSeq2500 system is of pico grams of concentration; thus this ensures that the difference of starting concentration is compensated.

Extra corrections 

-We have updated Table 1:

1) Cat breed: 2 animals were originally described as “unknown”. Owners have subsequently provided with breed information, leading to them being changed to DSH 

2) Age in years (median range) of some of the cats has been updated: 

Healthy: unchanged

PN: 7 (2-13)

SB/C: unchanged

UO: 5 (2-11)

CKD: 12 (8-17)

-Because a new Supplementary Table has been created, all Supplementary Table numbers have been updated accordingly. 

-Figure 3 has been improved by showing the cat urine samples with diamond shape in order to distinguish from the dog samples (round shape). Therefore the old figure 3 has been substituted by the new Figure 3.

-A title of one paragraph in the discussion section has been slightly modified: see lines 684-685:

From “Differentiating pyelonephritis from other urological conditions”

To “Differentiating cats with pyelonephritis from cats with other urological conditions”

-The nomenclature for the healthy cats has been homogenized along the text using “Control” cats.

-Small typos and grammar have been corrected along the text and tables. Also, few sentences have been added where needed.

---

## [Decision Letter · Decision Letter 1]

11 May 2022

PONE-D-22-02085R1Urinary microRNAome in healthy cats and cats with pyelonephritis or other urological conditionsPLOS ONE

Dear Dr. Cirera,

Thank you for submitting your manuscript to PLOS ONE. Both reviewers have recommended publication. However, please change "gender" for "sex" throughout the manuscript, as suggested by one of the reviewer. The manuscript will be accepted shortly after your resubmission.

We look forward to receiving your revised manuscript.

Kind regards,

Silvia Sabattini

Academic Editor

PLOS ONE

Journal Requirements:

Reviewers' comments:

Reviewer's Responses to Questions

**Comments to the Author**

1. If the authors have adequately addressed your comments raised in a previous round of review and you feel that this manuscript is now acceptable for publication, you may indicate that here to bypass the “Comments to the Author” section, enter your conflict of interest statement in the “Confidential to Editor” section, and submit your "Accept" recommendation.

Reviewer #1: All comments have been addressed

Reviewer #2: All comments have been addressed

2. Is the manuscript technically sound, and do the data support the conclusions?

Reviewer #1: Yes

Reviewer #2: Yes

3. Has the statistical analysis been performed appropriately and rigorously? 

Reviewer #1: Yes

Reviewer #2: Yes

4. Have the authors made all data underlying the findings in their manuscript fully available?

Reviewer #1: Yes

Reviewer #2: Yes

5. Is the manuscript presented in an intelligible fashion and written in standard English?

Reviewer #1: Yes

Reviewer #2: Yes

6. Review Comments to the Author

Reviewer #1: (No Response)

Reviewer #2: Review of the Research Article PONE-D-22-02085R1: "Urinary microRNAome in healthy cats and cats with pyelonephritis or other urological conditions."

Minor change: Please change "gender" for "sex" throughout the manuscript. Although widely used, gender can be used in human medicine as an identity; thus, it is preferable to use sex for animals.

Thank you for clarifying that all the samples are from whole urine for miRNA expression studies and for including valuable data on miRNA concentration; this will be valuable for researchers working with miRNA expression in urine.

As a note, I agree Bioanalyzer is also not ideal for urine samples, and I understand the choice of running NanoDrop. My suggestion is to use Qubit Fluorometer with kits specific for miRNA. This is just a note and not a recommendation for this manuscript.

This manuscript is very well written and scientifically sound, and the authors satisfactorily answered all reviewer questions; thus, my recommendation is to accept the manuscript for publication in PLOS ONE.

7. PLOS authors have the option to publish the peer review history of their article (what does this mean?). If published, this will include your full peer review and any attached files.

Reviewer #1: No

Reviewer #2: **Yes: **Andrea Pires dos Santos

---

## [Author Response · Author response to Decision Letter 1]

16 May 2022

The suggested change did not exist in the manuscript text or in any figure. So we don't need to resubmit a new version.

---

## [Editor Report · Decision Letter 2]

3 Jun 2022

Urinary microRNAome in healthy cats and cats with pyelonephritis or other urological conditions

PONE-D-22-02085R2

Dear Dr. Cirera,

We’re pleased to inform you that your manuscript has been judged scientifically suitable for publication and will be formally accepted for publication once it meets all outstanding technical requirements.

Kind regards,

Silvia Sabattini

Academic Editor

PLOS ONE
---

## [Editor Report · Acceptance letter]

12 Jul 2022

PONE-D-22-02085R2 

Urinary microRNAome in healthy cats and cats with pyelonephritis or other urological conditions 

Dear Dr. Cirera:

I'm pleased to inform you that your manuscript has been deemed suitable for publication in PLOS ONE. Congratulations! Your manuscript is now with our production department. 

Kind regards, 

on behalf of

Dr. Silvia Sabattini 

Academic Editor

PLOS ONE